# Efficiently Learning Drifting Halfspaces with Massart Noise

**Mingchen Ma** [1]  **Guyang Cao** [1]  **Jelena Diakonikolas** [1]  **Ilias Diakonikolas** [1]

## Abstract

We study the problem of learning a drifting concept in the presence of Massart noise. In this framework, an online learner has access to a history of independent samples whose labels are noisy versions of a target concept that may change from round to round. The goal is to output, in each round, a hypothesis with small prediction error. We study the complexity of this learning problem for the fundamental class of margin-separable linear classifiers (halfspaces). On the positive side, we give a computationally efficient learner achieving error $\eta + \tilde{O}(\Delta^{1/3}/\gamma)$, where $\eta$ upper bounds the Massart noise rate, $\Delta$ is the drift rate, and $\gamma$ is the margin. Interestingly, in the realizable setting, an adaptation of our techniques yields an efficient learner with an improved error rate over prior work. On the lower-bound side, we provide formal evidence of an information-computation tradeoff, strongly suggesting that our algorithm's performance is essentially optimal. Specifically, while the information-theoretically optimal error scales with $\Delta^{1/2}$, we prove that $\Delta^{1/3}$-scaling is unavoidable for low-degree polynomial tests, even in the special case of random classification noise.

## 1 Introduction

Statistical learning theory (Vapnik, 1998) typically focuses on settings in which the training set consists of i.i.d. samples from a fixed unknown distribution, and the performance of the trained model is evaluated on the same distribution. This modeling assumption does not capture a range of machine learning applications, where the training data is drawn from a non-stationary distribution that drifts over time, and the deployed model needs to perform well given data drawn from a different distribution. Examples include financial

prediction, analysis of consumer preferences, weather forecast, and more recently language models. Specifically, in language model training (Arakelyan et al., 2023), the models are trained over static snapshots of text collected from websites that are frequently changing. When the models are deployed, they need to work with timely news, facts, or programming APIs. To address such settings, we require learning models and associated algorithms that succeed in the presence of distribution drift.

A prototypical statistical model to formalize such non-static settings is that of binary classification with a drift (Helmbold et al., 1991; Helmbold & Long, 1994; Bartlett, 1992). For concreteness, we formally define this model below.

**Definition 1.1** (Learning under Distribution Drift). Let $\mathcal{C}$ be a hypothesis class of $\{\pm 1\}$-valued functions over some domain $X$. For $t = 1, 2, \ldots$, let $D^{(t)}$ be a distribution over $X \times \{\pm 1\}$ such that $d_{TV}(D^{(t)}, D^{(t+1)}) \leq \Delta$, where $d_{TV}$ is the total variation distance and $\Delta \in (0, 1)$ is the drift rate. The error of a hypothesis $h : X \to \{\pm 1\}$ with respect to $D^{(t)}$ is $\mathrm{err}^{(t)}(h) := \mathbf{Pr}_{(x^{(t)}, y^{(t)}) \sim D^{(t)}}[h(x^{(t)}) \neq y^{(t)}]$. A learning algorithm $\mathcal{A}$ is given a sequence of examples $\{(x^{(t)}, y^{(t)})\}$, for $t = 1, 2, \ldots$, where $(x^{(t)}, y^{(t)}) \sim D^{(t)}$, and works in an online fashion. Specifically, given $\{(x^{(i)}, y^{(i)})\}_{i=1}^{t-1}$, $\mathcal{A}$ outputs a hypothesis $\hat{h}^{(t)} : X \to \{\pm 1\}$ and then receives an example $(x^{(t)}, y^{(t)})$ drawn from $D^{(t)}$. The goal of the learner is to minimize $\mathrm{err}^{(t)}(\hat{h}^{(t)})$, as compared to $\mathrm{opt}_t := \min_{h \in \mathcal{C}} \mathrm{err}^{(t)}(h)$.

This model has been extensively studied over the past decades (see Section 1.2). Much of the early work focused on determining the optimal error rate for general hypothesis classes. Specifically, for any class $\mathcal{C}$ of VC-dimension $d$, (Helmbold et al., 1991; Bartlett, 1992; Helmbold & Long, 1994; Barve & Long, 1997; Long, 1999) establish the optimal error rate of $\Theta((d\Delta)^{1/2})$ for the realizable case (i.e., when the labels in each round are consistent with a function in $\mathcal{C}$) and the excess error rate of $\Theta((d\Delta)^{1/3})$ in the agnostic setting. A key idea underlying these works is the following: When $d_{TV}(D^{(t)}, D^{(t+1)})$ is small, by carefully selecting a number of past samples that balance the estimation error and the error due to the distribution drift, empirical risk minimization (ERM) over the selected sample gives a hypothesis with statistically optimal error.

Unfortunately, the aforementioned statistically optimal ap-

[1] Department of Computer Sciences, University of Wisconsin-Madison, Madison, USA. Correspondence to: Mingchen Ma <mingchen@cs.wisc.edu>.

*Proceedings of the 43$^{rd}$ International Conference on Machine Learning*, Seoul, South Korea. PMLR 306, 2026. Copyright 2026 by the author(s).

proach leads to an exponential time algorithm in general—even in the realizable setting, where each $x^{(t)}$ is labeled according to a hypothesis $h^{(t)} \in \mathcal{C}$. More broadly, despite extensive investigation, the *computational complexity* of learning with distribution drift remains poorly understood, even for basic hypothesis classes. Of course, if the non-drift version of the problem (corresponding to $\Delta = 0$) is computationally intractable, the hardness is inherited in the drift setting. Hence, it is natural to focus on concept classes and models of label noise for which algorithmic results are known in the non-drift setting.

Here we study the complexity of drift learning for the class of linear classifiers or halfspaces. A halfspace is any function of the form $h_w(x) = \text{sign}(w \cdot x)$, where $w$ is a weight vector and $\text{sign}(t)$ is equal to 1 for $t \geq 0$ and $-1$ otherwise. Let $\mathcal{H} := \{h_w(x) = \text{sign}(w \cdot x) \mid w \in \mathbb{S}^{d-1}\}$ denote the class of halfspaces on $\mathbb{R}^d$, where $\mathbb{S}^{d-1}$ is the unit ball.

Prior algorithmic results for drifting halfspaces are restricted to the realizable setting and either achieve highly suboptimal error rates (Helmbold & Long, 1994) or require very strong distributional assumptions (Crammer et al., 2010; Hanneke et al., 2015). Departing from this setup, we aim to develop *efficient*[1] algorithms under minimal distributional assumptions and in the presence of label noise. More specifically, our only assumption on the distribution of examples will be the existence of margin $\gamma > 0$ with respect to the target halfspace. Regarding the noise model, we assume that the labels in each round are corrupted by Massart noise (Massart & Nedelec, 2006), one of the classical semi-random noise models in the literature for which efficient learners are known in the non-drift setting (Diakonikolas et al., 2019)[2].

In the Massart model, the label of example $x$ can be flipped with probability $\eta(x) \leq \eta < 1/2$, independently across examples. The noise function $\eta(x)$ is assumed to be bounded and is *unknown* to the learner. The special case of Massart noise where $\eta(x) = \eta$ for all $x$ is known as Random Classification Noise (RCN) (Angluin & Laird, 1988).

In the considered drift setting, we allow a different source of Massart noise in each round. Specifically, let $h_w^{(t)}(x)$ be the target function (halfspace) and let $\eta_t(x) : \mathbb{S}^{d-1} \to [0, \eta]$. For $t \geq 1$, we say that a distribution $D^{(t)}$ is realized by $(h_w^{(t)}, \eta_t(x))$ if for every $x$, $\eta_t(x) = \mathbf{Pr}_{(x,y) \sim D^{(t)}}[h_w^{(t)}(x) \neq y \mid x]$. Under Definition 1.1, we will restrict every $D^{(t)}$ to be realized by some unknown $(h_w^{(t)}, \eta_t(x))$. (The case of RCN corresponds to $\eta_t(x) \equiv \eta_t \leq \eta$.)

## 1.1 Our Results

Our work provides the first results for drifting concept classes under Massart noise, in terms of both statistical and computational complexity. For statistical complexity, in Section 2, we show that for a general hypothesis class with VC dimension $d$, the statistically optimal excess error with Massart noise—achieved by an ERM—is $\tilde{\Theta}(\sqrt{d\Delta})$ (and $\tilde{\Theta}(\sqrt{\Delta}/\gamma)$ for $\gamma$-margin halfspaces)—as opposed to $\Theta((d\Delta)^{1/3})$, which applies to adversarial label noise.[3]

We then characterize the error that can be achieved by *efficient* learning algorithms for learning $\gamma$-margin drifting halfspaces with Massart noise. Our main algorithmic result is the following:

**Theorem 1.2** (Main Algorithmic Result). *Consider the problem of learning drifting $\gamma$-margin halfspaces with $\eta$-Massart noise. There is an algorithm $\mathcal{A}$ that runs in* $\text{poly}(d, 1/\gamma, 1/\Delta)$ *time and outputs a hypothesis $\hat{h}^{(T)}$ such that for any time step $T = \tilde{\Omega}(\Delta^{-2/3})$, with probability at least $9/10$, $\text{err}^{(T)}(\hat{h}^{(T)}) \leq \eta + \tilde{O}(\Delta^{1/3}/\gamma)$.*

To the best of our knowledge, this is the first non-trivial error guarantee achieved by an efficient learning algorithm in the presence of label noise. We note that, for the special case of RCN, an adaptation of our algorithm gives an error of $\text{opt}_t + \tilde{O}(\Delta^{1/3}/\gamma)$. As another important implication, we obtain an efficient algorithm with an even smaller error $\tilde{O}(\sqrt{\Delta}\gamma^{-3/2})$ for the realizable setting—improving upon the $\tilde{O}(\sqrt{\Delta}\gamma^{-2})$ error achieved by the algorithm in (Helmbold & Long, 1994).

Interestingly, the error achieved by our noise-tolerant drift algorithm scales with $\Delta^{1/3}$, which is qualitatively worse than the statistically optimal error (that scales with $\Delta^{1/2}$). We complement our upper bound by providing formal evidence that the scaling with $\Delta^{1/3}$ is unavoidable for efficient learning algorithms, even in the special case of RCN. Specifically, we establish a lower bound against the class of low-degree polynomial tests (Hopkins, 2018; Kunisky et al., 2019; Wein, 2025)—one of the most well-studied computational models in the context of statistical-computational tradeoffs that captures a broad class of learning algorithms. Specifically, we show:

**Theorem 1.3** (Informal Implication of Theorem 4.4). *Consider the problem of learning drifting $\gamma$-margin halfspaces with $(1/3)$-RCN. For $T \geq \gamma^{-1/6}\Delta^{-2/3}$, there is a family of instances such that under the low-degree hardness conjecture, there is no polynomial time algorithm that achieves error $\text{opt}_T + \Delta^{1/3}\gamma^{-1/6}$ error for every instance in the family.*

An intuitive interpretation of Theorem 1.3 is that efficient algorithms for our problem require excess error

---

[1]Here and henceforth, "efficient" refers to polynomial-time algorithms in parameters describing the problem $(d, 1/\gamma, 1/\Delta)$.

[2]In the agnostic model, known hardness results rule out efficient algorithms even with a margin assumption (Daniely, 2016).

[3]The "tilde" notation hides logarithmic factors in the argument.

$\Omega(\Delta^{1/3})$, matching the guarantee of our algorithm. Taken together, these results illustrate the following phenomenon for bounded label noise (Massart, RCN) settings in terms of scaling with drift $\Delta$: while information-theoretic rates place these label noise models in the same category as the realizable setting—scaling with $\Delta^{1/2}$—computationally efficient algorithms need to pay the higher $\Delta^{1/3}$ price—the information-theoretic limit of the agnostic setting.

## 1.2 Related Work

**Learning with Distribution Drift** As already discussed, the early line of work on learning drifting concepts focused on establishing optimal excess error bounds for a general hypothesis class with VC-dimension $d$, under the classical assumption that the TV-distance between consecutive distributions is upper bounded by $\Delta$ (Helmbold et al., 1991; Bartlett, 1992; Helmbold & Long, 1994; Barve & Long, 1997; Long, 1999). In recent years, variants of this problem have been studied (Hanneke & Yang, 2019; Han et al., 2024; Mazzetto & Upfal, 2023; Baby et al., 2025) to develop methods that do not depend on the prior knowledge of $\Delta$. Importantly, all algorithms in these works rely on ERM and are thus not computationally efficient in general.

The aforementioned progress notwithstanding, the computational complexity of learning with drift is poorly understood. Early work (Helmbold & Long, 1994) gives an LP-based efficient algorithm for realizable halfspaces on $\mathbb{R}^d$ with error $\tilde{O}(d\sqrt{\Delta})$ (a bound that is suboptimal by a $\sqrt{d}$ factor). More recent works focused on developing efficient learners under different error metrics (Mohri & Muñoz Medina, 2012) or under strong assumptions on the marginal distribution— namely that each $D_x^{(t)}$ is the a uniform distribution over the unit sphere. Specifically, (Crammer et al., 2010) gave an efficient algorithm with error $O((d\Delta)^{1/4})$ that was later improved to $O(\sqrt{d\Delta})$ by (Hanneke et al., 2015). To the best of our knowledge, no efficient algorithms under the distribution drift settings were known to tolerate label noise.

**Halfspace Learning with Label Noise** Designing efficient learners for halfspaces in the presence of label noise has been a central problem in learning theory that has seen substantial progress in the past decades. In the non-drift version of the problem, when the marginal distribution is sufficiently structured (e.g., uniform over the unit sphere or Gaussian), efficient agnostic learners that can tolerate adversarial label noise are known (Awasthi et al., 2017; Diakonikolas et al., 2018; 2022b; 2024). For more general distributions (e.g., in the presence of margin), known hardness results (Daniely, 2016) rule out polynomial-time agnostic learners. In the presence of Massart noise, recent work (Diakonikolas et al., 2019; Chen et al., 2020; Diakonikolas & Zarifis, 2024; Chandrasekaran et al., 2024; Kontonis et al., 2024) has developed efficient learners achieving error $\eta + \epsilon$, a guarantee

shown to be optimal for efficient algorithms (Diakonikolas & Kane, 2022; Nasser & Tiegel, 2022; Diakonikolas et al., 2022a).

**Low-Degree Polynomial Hardness** Proving lower bounds against restricted algorithmic families such as SQ algorithms (Kearns, 1998) or low-degree polynomial estimators (Hopkins, 2018; Brennan et al., 2021) is a popular method of studying the computational complexity of learning problems. Low-degree polynomial estimators (defined in Section 4) provide a natural computational model for our online learning setting. For recent progress on low-degree polynomial hardness, we refer the reader to (Wein, 2025).

## 1.3 Notation

For a halfspace $h_w(x) = \text{sign}(w \cdot x)$ and $x \in \mathbb{S}^{d-1}$, we define $\text{err}^{(i)}(w, x) := \mathbf{Pr}_{(x,y)\sim D^{(i)}}[h_w(x) \neq y \mid x]$. For $\gamma \in (0, 1)$, an example $x \in \mathbb{S}^{d-1}$ is said to have $\gamma$-margin with respect to a halfspace $h_w(x)$ if $|w \cdot x| \geq \gamma$. A distribution $D$ is said to have $\gamma$-margin with respect to $h_w(x)$ if $\mathbf{Pr}_{x\sim D_x}[|w \cdot x| < \gamma] = 0$. For $a, b \in \mathbb{R}$, we denote by $\phi(a, b) = \text{sign}(a)\text{sign}(b) \in \{\pm 1\}$. We use $O_\delta(\cdot)$ notation to hide a $\log(1/\delta)$ factor and use $\tilde{O}(\cdot)$ to hide the polylogarithmic factors in the argument inside the big-Oh.

## 2 Information-Theoretic Baselines under Massart Noise

The literature on learning drifting concepts with label noise predominantly focuses on the agnostic setting (adversarial label noise), where for every hypothesis class with VC dimension $d$, the optimal excess error is $\Theta\left((d\Delta)^{1/3}\right)$ (Long, 1999). However, agnostic learning is known to be computationally hard for natural hypothesis and distribution classes, even without distribution shift (i.e., for $\Delta = 0$). Since our focus in this work is on computationally efficient algorithms, we consider structured label noise, for which we establish different excess error baselines.

We first show that in the presence of Massart noise, a smaller excess error rate of $\tilde{O}(\sqrt{d\Delta})$ can be obtained (see Appendix B.1 for the proof). This result, summarized in the following theorem, establishes an *information theoretic* upper bound; however, the algorithm itself is not polynomial-time. We highlight that the stated upper bound applies to any Boolean concept class (not just halfspaces) and the general case of Massart noise.

**Theorem 2.1** (Information-Theoretic Upper Bound)**.** *Consider the problem of learning drifting binary concepts. Let $\mathcal{H} : X \rightarrow \{\pm 1\}$ be a hypothesis class with VC dimension $d$. Suppose that for every $i > 0$, $D^{(i)}$ is realized by $((h^*)^{(i)}, \eta_i(x))$ for some $(h^*)^{(i)} \in \mathcal{H}$ and $\eta_i$ that satisfies the $\eta$-Massart noise condition. There is an algorithm $\mathcal{A}$ such that for every $t = \tilde{\Omega}(d/((1 - 2\eta)\Delta)$, $\mathcal{A}$ outputs a*

hypothesis $\hat{h}^{(t)}$ *such that with probability at least* $9/10$ *the error of* $\hat{h}^{(t)}$ *is at most* $\mathrm{opt}_t + \tilde{O}((d\Delta/(1-2\eta))^{1/2})$.

On the lower bound side, we specialize to halfspaces with $\eta$-RCN (the more specialized the class, the stronger the lower bound). We give a matching $\Omega((d\Delta)^{1/2})$ bound on the excess error, as summarized in Theorem 2.2 (see Appendix B.2 for the proof). Combined with Theorem 2.1, this establishes the information-theoretic baseline of $\tilde{\Theta}((d\Delta)^{1/2})$ for the excess error under both the Massart noise and RCN, provided $1/2 - \eta$ can be treated as a positive constant (the primary case for which these noise models are studied).

**Theorem 2.2.** *For every* $T > 0$ *and* $\Delta \in (0,1)$, *there is a family of instances of learning halfspaces with* $\eta$-RCN *such that provided* $(1-2\eta)^3 > d\Delta$, *there is no learning algorithm* $\mathcal{A}$ *that can achieve error* $\mathrm{err}^{(T)}(\mathcal{A}) \leq \mathrm{opt}_T + o(\sqrt{d\Delta/(1-2\eta)})$, *with probability* $1/2$ *for every instance in the family.*

# 3 Efficiently Learning Drifted Halfspaces with Massart Noise

In this section, we establish Theorem 1.2.

Our main learning algorithm, Algorithm 1, works in epochs, where the $i^{\mathrm{th}}$ epoch corresponds to time steps $t \in \{(i-1)W + 1, \ldots, iW\}$. In epoch $i$, Algorithm 1 maintains a hypothesis $\hat{h}^{(i)}$ for prediction and collects a set of examples $S^{(i)}$ observed during this epoch. At the end of the epoch, Algorithm 1 updates the hypothesis $h^{(i)}$ using the collected dataset $S^{(i)}$ through a subroutine in Algorithm 2.

---

**Algorithm 1** DRIFTEDMASSART (Learning Drifting Halfspaces with Massart Noise)

---

1: **Input:** noise level $\eta \in (0, 1/2)$, margin parameter $\gamma \in (0,1)$, drift rate $\Delta > 0$
2: **Output:** A hypothesis $h$ for each round $t$.
3: Let $W = 2\lfloor \Delta^{-2/3} \log((\Delta)^{-1}) \rfloor, i = 1, \hat{h}^0(x) \equiv 1$
4: **for** $t = 1, 2, \ldots$ **do**
5:    Output hypothesis $\hat{h}^{(i-1)}$ and receive $(x^{(t)}, y^{(t)})$
6:    $S^{(i)} \leftarrow S^{(i)} \cup \{(x^{(t)}, y^{(t)})\}$
7:    **if** $t = iW$ **then**
8:      $\hat{h}^{(i)} \leftarrow$ DRIFTPERCEPTRON$(\eta, \gamma, S^{(i)}, \gamma/\sqrt{W})$,
     $S^{(i)} \leftarrow \emptyset, i \leftarrow i+1$

---

Algorithm 2 equally partitions $S^{(i)}$ into sets $S_1^{(i)}$ and $S_2^{(i)}$. Set $S_1^{(i)}$, consisting of the first half of the examples, is used to update the parameter vector $w^{(i)}$, while the second half of the examples in $S_2^{(i)}$ are used to select a good hypothesis halfspace $h_{w^{(i)}}$ from the trajectory of updates.

In each round, we feed Algorithm 2 an example $(x, y)$ and generate a "gradient" vector $g(w^{(i)}; x, y)$ based on the current vector $w^{(i)} \in \mathbb{S}^{d-1}$ and the example $(x, y)$. With such

---

**Algorithm 2** DRIFTPERCEPTRON (Subroutine for Learning a Single Halfspace over a Dataset)

---

1: **Input:** noise level $\eta \in (0, 1/2)$, margin parameter $\gamma \in (0,1)$, step size $\mu > 0$, a dataset $S = \{(x^{(i)}, y^{(i)})\}_{i=1}^{2m}$ of $2m$ labeled examples
2: **Output:** $\hat{h} = \mathrm{sign}(\hat{w} \cdot x) : \mathbb{R}^d \to \{\pm 1\}$
3: $w^{(0)} = e_1$
4: Split $S = S_1 \cup S_2$, where $S_1 = \{(x^{(i)}, y^{(i)})\}_{i=1}^m, S_2 = S \setminus S_1$
5: **for** $i = 1, 2, \ldots, m$ **do**
6:    $g(w^{(i)}; x, y) \leftarrow ((1 - 2\eta)\mathrm{sign}(w^{(i)} \cdot x) - y)x/\max\{|w^{(i)} \cdot x|, \gamma\}$
7:    $w^{(i+1)} \leftarrow \mathrm{proj}_{\mathbb{B}(1)}(w^{(i)} - \mu g_t(w^{(i)}, x, y))$,
   $\hat{h}_{i+1}(x) = \mathrm{sign}(w^{(i+1)} \cdot x)$
8: Return $\hat{h} = \mathrm{argmin}_{\hat{h}_i} \hat{\mathrm{err}}(\hat{h}_i)$, where $\hat{\mathrm{err}}(\hat{h}_i) = \frac{1}{|S_2|} \sum_{(x,y) \in S_2} \mathbb{1}\{\hat{h}_i(x) \neq y\}$.

---

a gradient vector, we modify $w^{(i)}$ by running a single step projected gradient descent with a carefully chosen step size. By standard regret analysis, we get the following fact, the proof of which is deferred to Appendix C.1.

*Fact* 3.1 (Progress Measure Measurement). Let $t > 0$ be any time step and let $S = \{(x^{(i)}, y^{(i)})\}_{i=1}^m$ be such that $(x^{(i)}, y^{(i)}) \sim D^{(t+i)}$. For $T \in [m]$ and $w^* \in \mathbb{S}^{d-1}$, Algorithm 2 satisfies

$$\frac{1}{T} \sum_{i=1}^T \mathop{\mathbf{E}}_{(x,y) \sim D^{(t+i)}}[g(w^{(i)}; x, y) \cdot (w^{(i)} - w^*)] \leq R(T, \mu, \gamma),$$

where $R(T, \mu, \gamma) = 1/(2T\mu) + 2\mu/\gamma^2 - \sum_{i=1}^T \xi_i/T, \xi_i := \left(g(w^{(i)}; x, y) - \mathbf{E}_{(x,y) \sim D^{(t+i)}} g(w^{(i)}; x, y)\right) \cdot (w^{(i)} - w^*)$.

Here, the term $1/(2T\mu) + 2\mu/\gamma^2$ can be interpreted as "optimization error" and the term $\sum_{i=1}^T \xi_i/T$ as "concentration error". We remark that the proof of Fact 3.1 only relies on the boundedness of the gradient used in each round. To make use of Fact 3.1 to analyze the performance of the halfspace $h_{w^{(i)}}$ in the update trajectory, we need to choose a suitable gradient $g(w; x, y)$ that can relate the expected regret term $\mathbf{E}_{(x,y) \sim D^{(t+i)}}[g(w^{(i)}; x, y) \cdot (w^{(i)} - w^*)]$ to the quantity of interest—the prediction error.

Our choice of the gradient vector is inspired by (Diakonikolas et al., 2025), which considers an online learning problem where labels are generated according to a *fixed* halfspace with Massart noise. The choice of $g(w^{(i)}; x, y)$ can be seen as the gradient of the leaky ReLU function $\ell_\eta(w; x, y) = ((1 - 2\eta)|yw \cdot x| + yw \cdot x)$ weighted by $\max\{|w^{(i)} \cdot x|, \gamma\}^{-1}$, where the chosen weight is used to balance penalizations for misclassified examples with different margins.

Although the choice of gradient is similar, the analysis from

prior work does not apply to the setting where the distribution drifts. The main difficulty is that under drifting distribution over the labeled examples, there is no single ground truth halfspace $h_{w^*}$ that can realize all distributions $D^{(i)}$, $i \geq 0$. Denote by $(w^*)^{(i)}$ the parameter vector of the halfspace that realizes $D^{(i)}$. When the marginal distribution is a uniform distribution over the unit sphere, then prior work (Crammer et al., 2010; Hanneke et al., 2015) shows that if $d_{TV}(D^{(i)}, D^{(i+1)})$ is small, then $\|(w^*)^{(i)} - (w^*)^{(i+1)}\|$ is also small, which makes it possible to treat $(w^*)^{(i)}$ as approximately unchanged. However, when such a distributional assumption is removed, $d_{TV}(D^{(i)}, D^{(i+1)})$ may be small, but the geometric distance between $(w^*)^{(i)}$ and $(w^*)^{(i+1)}$ can still be quite large. We bypass the difficulty of analyzing the change of $\|(w^*)^{(i)} - (w^*)^{(i+1)}\|$ by showing that despite $\|(w^*)^{(i)} - (w^*)^{(i+1)}\|$ being potentially large, the prediction error in each round can always be controlled by the expected regret term plus a drift-controlled error term. This technical approach is summarized in our main technical lemma, stated below.

**Lemma 3.2** (Regret to Error). *Let $i, m \in \mathbb{N}$ and let $w^* \in \mathbb{S}^{d-1}$ be a vector that realizes $D^{(i+m)}$. There exists a function $F_i(w^*, w, x) : \mathbb{S}^{d-1} \times \mathbb{S}^{d-1} \times \mathbb{S}^{d-1} \to \mathbb{R}$ such that for any $x \in \mathbb{S}^{d-1}$ and for any $w \in \mathbb{S}^{d-1}$,*

$$\mathop{\mathbf{E}}_{y \sim D^{(i)}|x}[g(w; x, y) \cdot (w - w^*)] \geq 2(\mathrm{err}^{(i)}(w, x) - \eta) - F_i.$$

*Additionally, $|F_i| \leq 1/\gamma$, and for every $w \in \mathbb{S}^{d-1}$, $\mathbf{E}_{x \sim D_x^{(i)}} F_i(w^*, w, x) = O(\Delta m/\gamma)$.*

We give an overview of the proof of Lemma 3.2, deferring the full details of the proof to Appendix C.2.

The proof of Lemma 3.2 is divided into two cases based on whether $|w \cdot x| > \gamma$. For each case $C$, we construct a bounded function $F_i^{(C)}(w^*, w, x)$ such that $\mathbf{E}_{y \sim D^{(i)}|x}[g(w; x, y) \cdot (w - w^*)] \geq 2(\mathrm{err}^{(i)}(w, x) - \eta) - F_i^{(C)}(w^*, w, x)$ holds deterministically for every $x \in \mathbb{S}^{d-1} \cap C$ and $\mathbf{E}_{x \sim D_x^{(i)}}[F_i^{(C)}(w^*, w, x)\mathbb{1}\{C\}] = O(\Delta m/\gamma)$. Given this construction, the desired function $F_i$ can be taken as the summation of $F_i^{(C)}(w^*, w, x)\mathbb{1}\{C\}$.

The central technical part of the proof is to show that the expectation of the constructed function $F_i$, which we call "drift error", is essentially controlled by the amount of distribution drift $m\Delta$. Our construction of $F_i$ enables us to relate the expectation to the change of the disagreement region $\{x \mid (w^* \cdot x)((w^*)^{(i)} \cdot x) < 0\}$ as well as the change of the noise rate $\eta_i(x) - \eta_{i+m}(x)$. We establish the following two claims in order to bound the expectation of $F_i^{(C)}$, whose proofs are deferred to Appendix C.2.3 and Appendix C.2.4.

*Claim* 3.3. For $i, m \in \mathbb{N}$, let $D^{(i)}, D^{(i+m)}$ be realized by $((w^*)^{(i)}, \eta_i(x))$ and $(w^*, \eta_{m+i}(x))$. Then $\mathbf{Pr}_{x \sim D_x^{(i)}}[\phi(w^* \cdot x, (w^*)^{(i)} \cdot x) = -1] = O(\frac{\Delta m}{1 - 2\eta})$.

*Claim* 3.4. For $i, m \in \mathbb{N}$, let $D^{(i)}, D^{(i+m)}$ be realized by $((w^*)^{(i)}, \eta_i(x))$ and $(w^*, \eta_{m+i}(x))$. Let $\mathcal{E} := \{x : \phi(w^* \cdot x, (w^*)^{(i)} \cdot x) = 1, \eta_i(x) \geq \eta_{m+i}(x)\}$. Then $\mathbf{E}_{x \sim D_x^{(i)}}[(\eta_i(x) - \eta_{m+i}(x)\mathbb{1}\{\mathcal{E}\}] = O(\Delta m)$.

Lemma 3.2 shows that the excess error of a hypothesis in each round can be bounded by its "average regret" plus a contribution due to the drift error. Thus, to prove Theorem 1.2, we only need to show that the "average regret" can be bounded by $O(\Delta^{1/3}/\gamma)$, while controlling the drift error contribution. By Fact 3.1 and Lemma 3.2, we know that we need to control the optimization error $1/(2T\mu) + 2\mu/\gamma^2$, the drift error $O(T\Delta/\gamma)$, and the concentration error $\sum_{i=1}^T \xi_i$. The first two errors are minimized by carefully choosing the epoch length $W$ as well as the update step size $\mu$, while the concentration term is controlled by the following lemma.

**Lemma 3.5.** *For $t, i > 0$ and $w^* \in \mathbb{S}^{d-1}$, define $\xi_i := \left(g(w^{(i)}; x, y) - \mathbf{E}_{(x,y) \sim D^{(t+i)}} g(w^{(i)}; x, y)\right) \cdot (w^{(i)} - w^*)$, $(x, y) \sim D^{(t+i)}$. For every $T > 0$, with probability at least $1 - \delta/2$, it holds $\sum_{i=1}^T \xi_i \leq \gamma^{-1}\sqrt{T}\log(1/\delta)$.*

*Proof of Theorem 1.2.* To prove the correctness of Algorithm 1, we show that for $t = \ell W$, with probability $1 - \delta$, $\mathrm{err}^{(t)}(h^{(\ell)}) \leq \eta + \tilde{O}(\Delta^{1/3}/\gamma)$. Suppose this is correct, then for $t' = \ell W + j$, $1 \leq j \leq W$, we have with probability at least $1 - \delta$,

$$\mathrm{err}^{(t')}(h^{(\ell)}) \leq \mathop{\mathbf{Pr}}_{(x,y) \sim D^{(iW)}}[h^{(\ell)}(x) \neq y] + j\Delta$$
$$\leq \eta + \tilde{O}(\Delta^{1/3}/\gamma).$$

To bound above the error of $\mathrm{err}^{(t)}(h^{(\ell)})$ for $t = \ell W$, we apply Fact 3.1 with $t = \ell W$ and $w^* \in \mathbb{S}^{d-1}$, the vector that realizes $D^{(t+T)}$ with $T = W/2$ and apply Lemma 3.2 with $i = t + j$ and $m = T - j$ for each $j \in [T]$. We have with probability at least 0.95

$$\frac{1}{T}\sum_{j=1}^T \left(\mathrm{err}^{(t+j)}(w^{(j)}) - \eta\right)$$
$$\leq \frac{1}{T}\sum_{j=1}^T \mathop{\mathbf{E}}_{D^{(t+j)}}[g(w^{(j)}; x, y) \cdot (w^{(j)} - w^*)] + O\left(\frac{T\Delta}{\gamma}\right)$$
$$\leq \frac{1}{2T\mu} + \frac{\mu}{2\gamma^2} - \frac{1}{T}\sum_{j=1}^T \xi_j + O(T\Delta/\gamma)$$
$$\leq 1/(\sqrt{T}\gamma) + O(T\Delta/\gamma) - \frac{1}{T}\sum_{j=1}^T \xi_j$$
$$\leq 1/(\sqrt{T}\gamma) + O(T\Delta/\gamma) + O(1/(\sqrt{T}\gamma)) = \tilde{O}(\Delta^{1/3}/\gamma).$$

Here, the third inequality follows from the choice $\mu = \gamma/\sqrt{T}$, the forth inequality follows from Lemma 3.5 and the last inequality follows from the choice of $T = \tilde{\Theta}(\Delta^{-2/3})$. This implies that with probability 0.95 there

is at least one $j \in [T]$ such that , $\text{err}^{(t+j)}(w^{(j)}) \leq \eta + \tilde{O}(\Delta^{1/3}/\gamma)$, as $\min_{j \in [T]} \left( \text{err}^{(t+j)}(w^{(j)}) - \eta \right) \leq \frac{1}{T} \sum_{j=1}^{T} \left( \text{err}^{(t+j)}(w^{(j)}) - \eta \right)$.

We next argue that we are able to select a good hypothesis by looking at the empirical error. For each $j \in [T]$, consider the halfspace $\hat{h}_j = h_{w^{(j)}}$. By the assumption on the drift rate, we have $\left| \text{err}^{((\ell+1)W)}(\hat{h}_j) - \text{err}^{(t+t')}(\hat{h}_j) \right| \leq \Delta W = \Delta^{1/3}$ for every $t' \in [W/2]$, which implies that

$$\left| \text{err}^{((\ell+1)W)}(\hat{h}_j) - \frac{2}{W} \sum_{\ell=1}^{W/2} \text{err}^{(t+t'+W/2)}(\hat{h}_j) \right| \leq \Delta W.$$

Recall the definition of $\hat{\text{err}}(\hat{h}_j)$ from Algorithm 2 as the empirical error of $\hat{h}_j$ over an independent set $S_2^{(i)}$. By Hoeffding's inequality, we have

$$\mathbf{Pr}\left[ \hat{\text{err}}(\hat{h}_j) - \frac{2}{W} \sum_{\ell=1}^{W/2} \text{err}^{(t+\ell+W/2)}(\hat{h}_j) > \Delta^{1/3} \right] \leq \frac{0.05}{W}.$$

Thus, by the union bound, with probability at least 0.95, we have $\left| \hat{\text{err}}(\hat{h}_j) - \text{err}^{((\ell+1)W)}(\hat{h}_j) \right| \leq \Delta^{1/3}$ for $j \in [T]$. Since with probability 0.95, there is some $j^* \in [T]$ such that $\text{err}^{(t+j^*)}(w^{(j^*)}) \leq \eta + \tilde{O}(\Delta^{1/3}/\gamma)$, we know that $\text{err}^{((\ell+1)W)}(w^{(j^*)}) \leq \eta + \tilde{O}(\Delta^{1/3}/\gamma)$. This implies that the selected hypothesis that minimizes $\hat{\text{err}}(h_j)$ must have $\text{err}^{((\ell+1)W)}(\hat{h}) \leq \eta + \tilde{O}(\Delta^{1/3}/\gamma)$, with probability 0.9. □

**Efficiently Learning Drifting Halfspaces with RCN** Under the Massart noise condition, even without distribution drift, it is computationally hard (Nasser & Tiegel, 2022) to output a hypothesis with error less than $\eta - o(1)$. For the special case of RCN, although $\text{opt}_t$ can change over time due to the distribution drift, we show that within each epoch, there is some $\eta^*$ such that for every time step $t$ within the epoch, $\text{opt}_t \leq \eta^* + O(\Delta W)$. Thus, by repeatedly running Algorithm 1 with different choices of $\eta$, we are able to get error $\text{opt}_t + \tilde{O}(\Delta^{1/3}/\gamma)$. Specifically, we show (see Appendix C.4 for the proof).

**Theorem 3.6** (Efficient Drift Learning with RCN). *Consider the problem of learning drifting halfspaces with $\eta$-RCN. There is an algorithm $\mathcal{A}$ such that for any time step $T = \tilde{\Omega}(\Delta^{-2/3})$, $\mathcal{A}$ runs in $\text{poly}(d, 1/\gamma, \Delta)$ time and outputs a hypothesis $\hat{h}^{(T)}$ such that with probability at least $9/10$, $\text{err}^{(T)}(\hat{h}^{(T)}) \leq \text{opt}_T + \tilde{O}(\Delta^{1/3}/\gamma)$.*

**Efficiently Learning Realizable Drifting Halfspaces** Another application of our algorithmic framework is an efficient algorithm for drifting halfspaces in the realizable setting with an improved error guarantee. When the marginal distribution is uniform over the unit sphere in $\mathbb{R}^d$, (Hanneke et al., 2015) shows that it is possible to efficiently achieve

an error of $\sqrt{d\Delta}$. However, when the strong distributional assumption is removed, the only known efficient learning algorithm with a provable error guarantee $d\sqrt{\Delta}$ is designed by (Helmbold & Long, 1994). Such an algorithm uses a simple observation that if the epoch length is $W = O(\Delta^{-1/2})$, then in the realizable setting, with constant probability, all observed examples within the epoch are consistent with a single halfspace and thus empirical risk minimization can be efficiently implemented. In the case of learning a $\gamma$-margin halfspace, such a trick only gives an error of $O(\sqrt{\Delta}\gamma^{-2})$ through a standard generalization analysis, because the sample size used to update the hypothesis is too small to achieve a good generalization error. To overcome such a difficulty, we increase the length of an epoch to $(\gamma\Delta)^{-1/2}$. Although within such a larger epoch, empirical risk minimization is no longer efficiently implementable, because the labels are no longer consistent with a single halfspace , our algorithm uses a carefully designed projected gradient descent to get a better generalization error $\tilde{O}(\sqrt{\Delta}\gamma^{-3/2})$.

**Theorem 3.7** (Efficient Drift Learning in Realizable Case). *Consider the problem of learning drifting halfspaces. There is an algorithm $\mathcal{A}$ such that for any time step $T = \tilde{\Omega}((\gamma\Delta)^{-1/2})$, $\mathcal{A}$ runs in $\text{poly}(d, 1/\gamma, \Delta)$ time and outputs a hypothesis $\hat{h}^{(T)}$ such that with probability at least $9/10$, $\text{err}^{(T)}(\hat{h}^{(T)}) = \tilde{O}(\sqrt{\Delta}\gamma^{-3/2})$.*

Compared to our Massart Algorithm 1, there are differences in the design and analysis of the learner in the realizable setting. Without label noise, we do not weight the gradient by $\max\{\left|w^{(i)} \cdot x\right|, \gamma\}^{-1}$ and use $g(w, x, y) \leftarrow (\text{sign}(w \cdot x) - y)x$ instead. Such a choice has the property that $\mathbf{E}_{(x,y) \sim D^{(i)}} \|g(w; x, y)\|^2 = \text{err}^{(i)}(h_w)$. This allows for a tighter regret bound as well as better control of drift error, and results in a better error guarantee of order $\Delta^{1/2}$ instead of $\Delta^{1/3}$. See Appendix C.5 for the formal details.

## 4 Low-Degree Polynomial Hardness

In this section, we establish our information-computation tradeoff for learning drifting halfspaces with RCN, thereby establishing Theorem 1.3.

To establish the hardness result, we first introduce the following testing problem, which we will show later can be solved by a learning algorithm with a small excess error.

**Definition 4.1** (Trajectory Testing Problem). A trajectory testing problem $\mathcal{B}(D^0, \mathcal{D})$ is defined by a distribution $D^{(0)}$ and a family of distributions $\mathcal{D}$ over $Z = (\mathbb{R}^d \times \{\pm 1\})^T$. An algorithm is given a sample $z = (z_1, \ldots, z_T) \sim D, z_i = (x_i, y_i), i \in [T]$, and is asked to distinguish whether $D = D^{(0)}$ or if $D$ is a distribution drawn uniformly from $\mathcal{D}$. In particular, this is the following hypothesis testing problem:

1. Null hypothesis $H_0$: All labeled examples $z_i$, $i \in [T]$,

are drawn from the same distribution: $D_i^{(0)} = D_j^{(0)}$, $\forall i, j \in [T]$. Furthermore, for each $i \in [T]$, $y_i$ is independent of $x_i$ and such that $\mathbf{Pr}_{D_i^{(0)}}(y_i = 1) = \eta \in (0, 1/2)$.

2. Alternative hypothesis $H_1$: Labeled examples satisfy $z_i \sim D_i$, where $D \in \mathcal{D}$ and each distribution $D_i$ corresponds to an instance of learning $\gamma$-margin halfspaces under RCN with noise rate $\eta_i \leq \eta \in (0, 1)$. Furthermore, for every $i \in [T]$, $d_{TV}(D_i, D_{i+1}) \leq \Delta$.

To establish our computational hardness result, we need to formally define the computational model. We restrict the learning algorithms to the family of low-degree polynomial estimators, one of the most commonly used algorithmic families in statistics and machine learning literature (Wein, 2025). More formally, Let $z = (z_1, \ldots, z_m) \in \mathbb{R}^{d \times m}$ be an input sample. A degree $k$ polynomial algorithm $\mathcal{A}$ specifies a function $f(z) = (f_1(z), \ldots, f_d(z)) : \mathbb{R}^{d \times m} \to \mathbb{R}^d$ such that for $i \in [d]$, $f_i(z)$ is a polynomial over $z$ with degree at most $k$. We say $f$ has a samplewise degree $(\ell, k)$ if $f$ can be written as a linear combination of monomials such that each monomial has a degree $\ell$ at each $z_i$ and has non-zero degree for at most $k$ examples.

**Definition 4.2** (Low-Degree Polynomial Test under Concept Drift). Under Definition 4.1, let $p : \mathbb{R}^{(d+1) \times T} \to \mathbb{R}$ be a polynomial over $Z$. A low degree polynomial test using $p$ with threshold $t$ is an algorithm such that given a sample $z$, it returns $H_0$ if $p(z) > t$. Furthermore, we say the polynomial $p$ is an $\epsilon$-distinguisher for Definition 4.1 if

$$\left| \mathop{\mathbf{E}}_{z \sim D^{(0)}} p(z) - \mathop{\mathbf{E}}_{D \sim \mathcal{D}} \mathop{\mathbf{E}}_{z \sim D} p(z) \right| \geq \epsilon \sqrt{\mathbf{Var}_{z \sim D^{(0)}} p(z)}.$$

Intuitively, at a given time step $T$, an algorithm is given the history of observed examples and is allowed to compute any polynomial function over the trajectory of the examples. If a polynomial $p$ is not an $\epsilon$-distinguisher for the testing problem, then for a trajectory $z$ sampled under $H_0$ or $H_1$, the result of $p(z)$ will have difference about $\epsilon$ with constant probability and thus is not reliable to distinguish the two cases. In particular, evaluating a *general* degree $k$ polynomial over $\mathbb{R}^{(d+1) \times T}$ requires time $(dT)^k$, which implies that if we can rule out super-constant degree polynomials as 1-distinguishers, we can provide evidence of the computational hardness of the problem.

**Hardness Construction** We remark that without any restrictions on the parameters of the problem, in the trajectory testing problem defined in Definition 4.1, the TV distance between $H_0$ and $H_1$ could be arbitrarily small, in which case the problem would be hard to solve even information-theoretically. Here we construct $D^{(0)}$ and $\mathcal{D}$ such that any algorithm that has a good error guarantee under the drift model can be used to solve the testing problem. Specifically, we consider examples supported over $\{\pm 1\}^d$, for $\gamma = \Theta(1/d)$ and force the marginal distribution to be the uniform distribution over the hypercube.

**Definition 4.3** (Hard Instance). Let $T \in \mathbb{N}^+$ be any sufficiently large number, let $\Delta \in (0, 1)$, $\gamma < \log^{-1}(1/\Delta)$, and let $d = \Theta(1/\gamma)$ be an odd number. We construct an instance of trajectory testing as follows.

1. Null $H_0$ : For each $i \in [T]$, $D_x^{(0)}$ is the uniform distribution over $\{\pm 1\}^d$ and $\mathbf{Pr}_{D_i^{(0)}}[y(x) = 1] = 1 - \eta$.

2. Alternative hypothesis $H_1$ : Let $S$ be a set of $2^{d^c}$ vectors in $\{\pm 1\}^d$ such that for every $v, u \in S$ and $v \neq u$, $|v \cdot u| \leq d^{1/2+c}$, $c \in (0, 1/2)$. For every $v \in S$, we define the distribution $D^{(v)}$ as follows. For $i \leq T - m$, $(D_i^{(v)})_x$ is the uniform distribution over $\{\pm 1\}^d$ and $\mathbf{Pr}_{D_i^{(v)}}[y(x) = 1] = 1 - \eta$. For $i = T - m + j$, $j \in [m]$, we define the ground truth halfspace to be $h_i^{(v)} = \mathrm{sign}(v \cdot x + t_i)$, where $\mathbf{Pr}_{D_i^{(v)}}[h_i^{(v)}(x) = -1] = \Delta j$ and $\mathbf{Pr}_{D_i^{(v)}}[h_i^{(v)}(x) \neq y(x) \mid x] = (\eta - \Delta j)/(1 - 2\Delta j) < \eta$. Here, $m = \gamma^{-1/6} \Delta^{-2/3}$.

We remark that in the construction of the hard instance, we consider halfspaces with thresholds instead of homogeneous halfspaces. This is for the purpose of simplifying the notation. Since the threshold $|t| = O(d)$, we are able to add $O(d)$ artificial dimensions to represent $(w, t) \in \mathbb{R}^{d+1}$ as a single vector $\bar{w} \in \mathbb{R}^{2d}$ such that $\|\bar{w}\|_\infty < 1$. This will make the halfspaces homogeneous and keep the margin $\gamma = \Theta(1/d)$.

In Appendix D.1, we show that for a reasonable $\eta$ (e.g., $\eta = 1/3$) the hard instance from Definition 4.3 is a valid instance of the trajectory testing problem. In Appendix D.5, we show that any learner that achieves error $\mathrm{opt}_i + \Delta^{1/3} \gamma^{-1/6}$ can be used efficiently to solve the testing problem. Thus, a low-degree polynomial hardness of the trajectory testing problem of Definition 4.3 can be used to give a hardness to an algorithm with an excess error $o(\Delta^{1/3})$. Thus, our main hardness result is as follows.

**Theorem 4.4.** *Let $c \in (0, 1/2)$ be a constant. For $\eta = 1/3$ and $\Delta > 2^{-1/\gamma^c}$, there is no polynomial $p(z)$ with degree less than $O(\gamma^{-c/4})$ that is a 1-distinguisher for the trajectory testing problem defined in Definition 4.1, with instance constructed by Definition 4.3.*

Intuitively, the hard instance constructed in Definition 4.3 requires us to distinguish two cases with RCN under a uniform distribution over the hypercube in $\mathbb{R}^d$. In the first case, the ground truth label is always $+1$. In the second case, the ground truth label remains $+1$ until time step $T - m$. But from time step $T - m + 1$, the distribution starts drifting along a random direction $v$. That is to say, we fix the direction of the normal vector of the target halfspace to be $v$, but gradually increase the fraction of examples with ground truth label $-1$ from 0 to $\Theta(\Delta m)$. Such a construction implies that no matter how large $T$ is, the only examples that

play a role are the last $m$ ones. However, due to the presence of RCN, the variance of the distributions of the labels is too large to be distinguished by a low-degree polynomial over the last $m$ samples.

We make use of linear algebraic tools to formalize this intuition. Let $D^{(0)}, D$ be distributions over $\mathbb{R}^d$. We denote by $\bar{D} := D(x)/D^{(0)}(x) : \mathbb{R}^d \to \mathbb{R}$ the relative density function between $D$ and $D^{(0)}$. Let $f, g : \mathbb{R}^d \to \mathbb{R}$ be two functions. We define the inner product between $f, g$ with respect to $D^{(0)}$ as $\langle f, g \rangle_{D^{(0)}} = \mathbf{E}_{x \sim D^{(0)}} f(x)g(x)$ and define $\|f\|_{D^{(0)}}^2 = \langle f, f \rangle_{D^{(0)}}$. For a function $f : \mathbb{R}^{d \times m} \to \mathbb{R}$, we denote by $f^{\leq k}$ the projection of $f$ onto the subspace of polynomials with degree at most $k$ with respect to $D^{(0)}$ and denote by $f^{\leq (\ell,k)}$ the projection of $f$ onto the subspace of polynomials with sample degree at most $(\ell, k)$ with respect to the inner product $\langle \cdot, \cdot \rangle_{D^{(0)}}$.

*Fact* 4.5 (One-sample version of Fact 2.1 and Fact 2.2 in (Brennan et al., 2021)). Let $\mathcal{C}_{\ell,k}$ be the linear subspace of polynomials over $z = (z_1, \ldots, z_T)$ with degree at most $\ell$ in each $z_i$ and with each monomial having a nonzero degree in at most $k$ of the $z_i$s. Then

$$\max_{p \in \mathcal{C}_{\ell,k}, \mathbf{E}_{D^{(0)}} p^2(z) \leq 1} \left( \mathbf{E}_{z \sim D^{(0)}} p(z) - \mathbf{E}_{D \sim \mathcal{D}} \mathbf{E}_{z \sim D} p(z) \right)$$
$$= \left\| \mathbf{E}_{D \sim \mathcal{D}} (\bar{D}^{\leq \ell,k}) - 1 \right\|_{D^{(0)}}.$$

Notice that if $\left\| \mathbf{E}_{D \sim \mathcal{D}}(\bar{D}^{\leq \ell,k}) - 1 \right\|_{D^{(0)}} \leq \alpha$, then there is no polynomial $p(z)$ in $\mathcal{C}_{\ell,k}$ that can distinguish $D^{(0)}$ from $\mathcal{D}$ with an advantage $\alpha$. Notice that $\mathcal{C}_{\infty,k}$ contains the family of polynomials of degree $k$. Thus, to prove Theorem 4.4, our goal is to show

$$\left\| \mathbf{E}_{D \sim \mathcal{D}} (\bar{D}^{\leq \infty,k}) - 1 \right\|_{D^{(0)}}^2$$
$$= \mathbf{E}_{D^{(u)}, D^{(v)} \sim \mathcal{D}} [\langle (\overline{D^{(u)}})^{\leq \infty,k}), (\overline{D^{(v)}})^{\leq \infty,k}) \rangle_{D^{(0)}}] - 1 \leq 1.$$

To do this, we first show that for every $u, v \in \{\pm 1\}^d$, $\langle (\overline{D^{(u)}})^{\leq \infty,k}), (\overline{D^{(v)}})^{\leq \infty,k}) \rangle_{D^{(0)}} - 1$ only depends on the last $m$ time steps in the trajectory.

**Lemma 4.6.** *Under the construction of Definition 4.3, we have for every $\ell, k$,*

$$\langle (\overline{D^{(u)}})^{\leq \ell,k}), (\overline{D^{(v)}})^{\leq \ell,k}) \rangle_{D^{(0)}} - 1$$
$$= \sum_{A \subseteq [m], A \neq \emptyset, |A| \leq k} \prod_{j \in [A]} I_j(u, v),$$

$$I_j(u, v) = \langle (D_{T-m+j}^{(u)})^{\leq \ell}(z_i)), (D_{T-m+j}^{(v)})^{\leq \ell}(z_i) \rangle_{D_i^{(0)}} - 1.$$

We defer the proof of Lemma 4.6 to Appendix D.2. With Lemma 4.6, it suffices to bound $I_j(u, v) = \langle (D_{T-m+j}^{(u)})^{\leq \ell}(z_i)), (D_{T-m+j}^{(v)})^{\leq \ell}(z_i) \rangle_{D_i^{(0)}} - 1$.

**Lemma 4.7.** *Let $c \in (0, 1/2)$ be a constant. For $i = T - m + j, j \in [m]$ and $v, u \in \{\pm 1\}^d$ such that $|v \cdot u| \leq d^{1-c}$,*

$$\langle D_{T-m+j}^{(u)}(z_i), D_{T-m+j}^{(v)}(z_i) \rangle_{D_i^{(0)}} - 1 \leq \tilde{O}(d^{-(1/2-c)}(\Delta j)^2)$$

$$\langle D_{T-m+j}^{(u)}(z_i), D_{T-m+j}^{(u)}(z_i) \rangle_{D_i^{(0)}} - 1 \leq O(\Delta j).$$

We defer the proof of Lemma 4.7 to Appendix D.3. The proof of Theorem 4.4 now follows from Lemma 4.6 and Lemma 4.7. In particular, for any $A \subseteq [m]$ with $|A| \leq \gamma^{-c/4}$, the choice of $m = \gamma^{-1/6}\Delta^{-2/3}$ enables us to show

$$\mathbf{E}_{D^{(u)}, D^{(v)} \sim \mathcal{D}} \prod_{j \in [A]} I_j(u, v) \leq (1/(100m))^{|A|},$$

which implies for $k = \gamma^{-c/4}, c \in (0, 1/2)$,

$$\left\| \mathbf{E}_{D \sim \mathcal{D}} (\bar{D}^{\leq \infty,k}) - 1 \right\|_{D^{(0)}}^2$$
$$= \mathbf{E}_{D^{(u)}, D^{(v)} \sim \mathcal{D}} \langle (D^{(u)})^{\leq \infty,k}), (D^{(v)})^{\leq \infty,k}) \rangle_{D^{(0)}} - 1$$
$$= \sum_{A \subseteq [m], A \neq \emptyset, |A| \leq k} \mathbf{E}_{D^{(u)}, D^{(v)} \sim \mathcal{D}} \prod_{j \in [A]} I_j(u, v)$$
$$\leq \sum_{\ell=1}^{k} \binom{m}{\ell} (1/(100m))^\ell \leq 1.$$

Due to the space limitations, we defer the full proof of Theorem 4.4 to Appendix D.4.

## 5 Conclusion

This paper provides information-theoretic rates and the first polynomial-time algorithms for learning drifting halfspaces with structured classes of noise, such as the semi-random Massart noise and its special case of random classification noise, including the noiseless (realizable) model. Additionally, we prove lower bounds applying to polynomial-time algorithms within the broad class of low-degree polynomial tests, providing strong evidence of an information-computation tradeoff for this problem.

A number of interesting questions for future work remain. For a concrete example, while polynomial-time algorithms with nontrivially bounded error are out of reach for general distributional families of covariates and adversarial labels, learning halfspaces under structured covariate distributions (e.g., logconcave distributions) and arbitrary labels is possible with polynomial-time algorithms for the settings with no distribution drift. Are such results possible in the drifting concept model? More broadly, it would be interesting to explore the computational aspects of learning with drift for other natural hypothesis classes, such as intersections of halfspaces and neural networks.

## Impact Statement

This paper presents work whose goal is to advance the field of Machine Learning. There are many potential societal consequences of our work, none which we feel must be specifically highlighted here.

## 6 Acknowledgments

Mingchen Ma was supported by NSF Award CCF-2144298 (CAREER). Ilias Diakonikolas was supported by the H.I. Romnes Faculty Fellowship, and the ONR (N00014-25-1-2268). Jelena Diakonikolas was supported in part by the Air Force Office of Scientific Research under award number FA9550-24-1-0076, NSF CAREER Award CCF-2440563, and NSF MFAI Award DMS-2502282. Guyang Cao was supported by NSF CAREER Award CCF-2440563. Any opinions, findings, conclusions, or recommendations expressed in this material are those of the authors and do not necessarily reflect the views of the U.S. Department of Defense.

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

## Supplementary Material

The structure of this supplementary material is the following: In Appendix A, we record some standard concentration inequalities. In Appendix B, we provide the omitted proofs from Section 2, establishing information-theoretic baselines for learning under Massart noise. In Appendix C, we provide omitted details from Section 3, providing efficient learning algorithms under distribution drift. In Appendix D, we give the proofs omitted from Section 4.

## A    Useful Concentration Inequalities

Our technical sections will make use of the following concentration inequalities.

*Fact* A.1 (Hoeffding's Inequality). Let $X_1, \ldots, X_n$ be $n$ independent random variables such that $X_i \in [a_i, b_i]$. Let $S_n = \sum_{i=1}^n X_i$. Then for $t > 0$, $\mathbf{Pr}\left[|S_n - \mathbf{E}\, S_n| > t\right] \le \exp(-2t^2 / \sum_{i=1}^n (a_i - b_i)^2)$.

*Fact* A.2 (Bernstein's Inequality). Let $X_1, \ldots, X_n$ be independent random variables such that $\mathbf{E}\, X_i = 0$ and $|X_i| \le M$. Let $\sigma^2 := \sum_{i=1}^n \mathbf{E}[X_i^2]$. Then for any $t > 0$, $\mathbf{Pr}(\sum_{i=1}^n X_i \ge t) \le \exp\left(-\frac{t^2}{2\sigma^2 + \frac{2}{3}Mt}\right)$.

*Fact* A.3 (Talagrand's Concentration Inequality). Let $X_1, \ldots, X_n$ be independent random variables taking values in a measurable space $\mathcal{X}$. Let $\mathcal{F}$ be a countable class of measurable functions $f : \mathcal{X} \to [-b, b]$ such that $\mathbb{E}[f(X_i)] = 0$, for all $f \in \mathcal{F}$, $i \in [n]$. Define $Z := \sup_{f \in \mathcal{F}} \sum_{i=1}^n f(X_i)$, and $\sigma^2 := \sup_{f \in \mathcal{F}} \sum_{i=1}^n \mathbb{E}[f(X_i)^2]$, then for all $t > 0$, $\mathbf{Pr}(Z \ge \mathbb{E}[Z] + t) \le \exp\left(-\frac{t^2}{2\sigma^2 + 2bt}\right)$.

*Fact* A.4 (Dudley's Inequality). Let $(\mathcal{F}, \rho)$ be a metric space and let $\{X_f : f \in \mathcal{F}\}$ be a centered stochastic process (i.e., $\mathbf{E}\, X_f = 0$ for all $f$). Assume the process has sub-Gaussian increments with respect to $\rho$, then $\mathbf{E}\left[\sup_{f \in \mathcal{F}} X_f\right] \le \int_0^{\text{diam}(\mathcal{F}, \rho)} \sqrt{\log \mathcal{N}(\epsilon, \mathcal{F}, \rho)}\, d\epsilon$, where $\mathcal{N}(\epsilon, \mathcal{F}, \rho)$ is the covering number.

## B    Omitted Proofs from Section 2

### B.1    Proof of Theorem 2.1

For convenience, we restate the theorem below.

**Theorem B.1** (Restatement of Theorem 2.1). *Consider the problem of learning under drifting concepts. Let $H : X \to \{\pm 1\}$ be a hypothesis class with VC dimension $d$. There is an algorithm $\mathcal{A}$ such that if for every $i > 0$, $D^{(i)}$ is realized by $((h^*)^{(i)}, \eta_i(x))$ for $(h^*)^{(i)}$ and $\eta_i$ satisfies the $\eta$-Massart noise condition, then for every $t > \Omega(d/((1 - 2\eta)\Delta))$, $\mathcal{A}$ outputs a hypothesis $\hat{h}^{(t)}$ with error at most $\text{opt}_t + \tilde{O}((d\Delta/(1 - 2\eta))^{1/2})$.*

The algorithm we will analyze, Algorithm 3, works in epochs, where the $i + 1^{\text{th}}$ epoch corresponds to time steps $t \in \{iW + 1, \ldots, (i+1)W\}$. In epoch $i$, Algorithm 3 maintains a hypothesis $\hat{h}^{(i)}$ for prediction and collects a set of examples $S^{(i)}$ observed during this epoch. At the end of the epoch, Algorithm 3 updates the hypothesis $h^{(i)}$ using the collected dataset $S^{(i)}$ by running a empirical risk minimization over $S^{(i)}$.

---

**Algorithm 3** DRIFTEDERM(Learning Drifting Concept with Bounded Noise)

---

1: **Input:** $\mathcal{H}$ : hypothesis class, $\Delta > 0$ drift rate, $\eta$: noise rate
2: **Output:** For each time step $t \ge 1$, a hypothesis $h : X \to \{\pm 1\}$
3: Let $W = \tilde{\Theta}(\sqrt{d\Delta^{-1}(1 - 2\eta)^{-1}}), i = 1, \hat{h}^{(0)}(x) \equiv 1$
4: **for** $t = 1, 2, \ldots$ **do**
5:     Output hypothesis $\hat{h}^{(i-1)}$ and receive $(x^{(t)}, y^{(t)})$.
6:     $S^{(i)} \leftarrow S^{(i)} \cup \{(x^{(t)}, y^{(t)})\}$
7:     **if** $t = iW$ **then**
8:         $\hat{h}^{(i)} \leftarrow \operatorname{argmin}_{h \in \mathcal{H}} \widehat{\text{err}}(h) := W^{-1} \sum_{(x,y) \in S^{(i)}} \mathbb{1}\{h(x) \ne y\}, S^{(i)} \leftarrow \emptyset$
9:         $i \leftarrow i + 1$

---

*Proof of Theorem 2.1.* Consider the hypotheses constructed by Algorithm 3. The main part of the proof is to argue that for $T = iW$, $\text{err}^{(T)}(\hat{h}^{(i)}) \le \text{opt}_T + \tilde{O}((d\Delta/(1 - 2\eta))^{1/2})$. Suppose this is correct, then for $t' = iW + j, 1 \le j \le W$, we

have

$$
\begin{aligned}
\mathrm{err}^{(t')}(\hat{h}^{(i)}) &= \Pr_{(x,y)\sim D^{(t')}}(\hat{h}^{(i)}(x) \neq y)\\
&\leq \Pr_{(x,y)\sim D^{(iW)}}(\hat{h}^{(i)}(x) \neq y) + j\Delta\\
&\leq \mathrm{opt}_{iW} + \tilde{O}((d\Delta/(1-2\eta))^{1/2}) + j\Delta\\
&\leq \mathrm{opt}_{t'} + \tilde{O}((d\Delta/(1-2\eta))^{1/2}) + 2j\Delta\\
&= \mathrm{opt}_{t'} + \tilde{O}((d\Delta/(1-2\eta))^{1/2}),
\end{aligned}
$$

where the first inequality comes from the drifting model assumption, the second inequality comes from assuming that for $T = iW$ (to be argued in the rest of the proof), $\mathrm{err}^{(T)}(\hat{h}^{(i)}) \leq \mathrm{opt}_T + \tilde{O}((d\Delta/(1-2\eta))^{1/2})$, and the third inequality again uses the bounded drift assumption.

In the rest of the proof, we argue that $\mathrm{err}^{(T)}(\hat{h}^{(i)}) \leq \mathrm{opt}_T + \tilde{O}((d\Delta/(1-2\eta))^{1/2})$ for $T = iW$.

For $i \geq 1$, let $T = iW$ and let $h^{(T)}$ be the hypothesis that realizes $D^{(T)}$. For each $t_j = (i-1)W + j$ and $h \in \mathcal{H}$, define $\mathrm{diff}_j(h) := \mathbb{1}\{h(x^{(t_j)}) \neq y^{(t_j)}\} - \mathbb{1}\{h^{(T)}(x^{(t_j)}) \neq y^{(t_j)}\}$. This gives

$$
\frac{1}{W}\sum_{j=1}^{W}\mathrm{diff}_j(h) = \hat{\mathrm{err}}(h) - \hat{\mathrm{err}}(h^{(T)}), \tag{1}
$$

and thus $W^{-1}\sum_{j=1}^{W}\mathrm{diff}_j(\hat{h}^{(i)}) \leq 0$ (by the definition of $\hat{h}^{(i)}$ in Algorithm 3).

Take expectations for each time step $t$ on both sides of (1), we get

$$
\begin{aligned}
&\mathbf{E}\left[\frac{1}{W}\sum_{j=1}^{W}\mathrm{diff}_j(h)\right]\\
&= \mathbf{E}[\hat{\mathrm{err}}(h)] - \mathbf{E}[\hat{\mathrm{err}}(h^{(T)})]\\
&= \frac{1}{W}\sum_{t=(i-1)W+1}^{T}\Pr_{(x^{(t)},y^{(t)})\sim D^{(t)}}(h(x^{(t)}) \neq y^{(t)}) - \frac{1}{W}\sum_{t=(i-1)W+1}^{T}\Pr_{(x^{(t)},y^{(t)})\sim D^{(t)}}(h^{(T)}(x^{(t)}) \neq y^{(t)}).
\end{aligned}
$$

By the drifting assumption and triangle inequality, we have $d_{\mathrm{TV}}(D^{(t)}, D^{(T)}) \leq W\Delta$ for any $t = (i-1)W + j$, $j \in [W]$. This implies that for any $h \in \mathcal{H}$, we have

$$
\begin{aligned}
&\left|\frac{1}{W}\sum_{t=(i-1)W+1}^{T}\Pr_{(x^{(t)},y^{(t)})\sim D^{(t)}}(h(x^{(t)}) \neq y^{(t)}) - \Pr_{(x^{(T)},y^{(T)})\sim D^{(T)}}(h(x^{(T)}) \neq y^{(T)})\right|\\
&\leq \frac{1}{W}\sum_{t=(i-1)W+1}^{T}\left|\Pr_{(x^{(t)},y^{(t)})\sim D^{(t)}}(h(x^{(t)}) \neq y^{(t)}) - \Pr_{(x^{(T)},y^{(T)})\sim D^{(T)}}(h(x^{(T)}) \neq y^{(T)})\right|\\
&\leq W\Delta. \tag{2}
\end{aligned}
$$

Apply (2) to $h$ and $h^{(T)}$, we have

$$
\left|\mathrm{err}^{(T)}(h) - \mathrm{err}^{(T)}(h^{(T)}) - \mathbf{E}\left[\frac{1}{W}\sum_{j=1}^{W}\mathrm{diff}_j(h)\right]\right| \leq 2W\Delta.
$$

By (1), we have $\frac{1}{W} \sum_{j=1}^{W} \mathrm{diff}_j(\hat{h}^{(i)}) \leq 0$, which means

$$\mathrm{err}^{(T)}(\hat{h}^{(i)}) - \mathrm{err}^{(T)}(h^{(T)})$$

$$\leq 2W\Delta + \mathbf{E}\left[\frac{1}{W} \sum_{t=(i-1)W+1}^{T} \mathrm{diff}_j(\hat{h}^{(i)})\right] - \frac{1}{W} \sum_{t=(i-1)W+1}^{T} \mathrm{diff}_j(\hat{h}^{(i)})$$

$$\leq 2W\Delta + \left|\frac{1}{W} \sum_{j=1}^{W} \mathrm{diff}_j(\hat{h}^{(i)}) - \mathbf{E}\left[\frac{1}{W} \sum_{j=1}^{W} \mathrm{diff}_j(\hat{h}^{(i)})\right]\right|. \tag{3}$$

To get the desired error rate, we will next show that with high probability, for every $h \in \mathcal{H}$, $\frac{1}{W} \sum_{j=1}^{W} \mathrm{diff}_j(h)$ is close to its expectation. Denote by $\sigma_j^2(h) = \mathbf{E}[\mathrm{diff}_j^2(h)] = \mathbf{Pr}_{(x^{(t)}, y^{(t)}) \sim D^{(t)}}[h(x^{(t)}) \neq h^{(T)}(x^{(t)})]$ and $V(h) := W^{-1} \sum_{j=1}^{W} \sigma_j^2(h)$. We first develop a $V(h)$-dependence bound for all $h \in H$.

For $r \in (0, 1)$, we denote by $H(r) := \{h \in \mathcal{H} \mid V(h) \leq r\}$. Notice that for every $h \in H(r)$, we have

$$W^{-1} \sum_{j=1}^{W} \mathbf{E}(\mathrm{diff}_j(h) - \mathbf{E}\,\mathrm{diff}_j(h))^2 \leq r.$$

Define random variable $Z(r) := \sup_{h \in H(r)} \left|W^{-1} \sum_{j=1}^{W} \mathrm{diff}_j(h) - W^{-1} \mathbf{E}\left[\sum_{j=1}^{W} \mathrm{diff}_j(h)\right]\right|$. By Talagrand's inequality (Talagrand, 1996), for any $\delta_r \in (0, 1)$, we have with probability $1 - \delta_r$

$$Z(r) \leq \mathbf{E}\,Z(r) + \sqrt{\frac{2r\log(1/\delta_r)}{W}} + \frac{3\log(1/\delta_r)}{W}.$$

It remains to upper bound $\mathbf{E}\,Z(r)$. By Theorem 8 in (Bartlett & Mendelson, 2002), we have

$$\mathbf{E}\,Z(r) \leq 2\mathop{\mathbf{E}}_{x,y}\mathop{\mathbf{E}}_{s} \sup_{h \in V(r)} W^{-1} \sum_{j=1}^{W} s_j \mathrm{diff}_j(h),$$

where $s \in \{\pm 1\}^W$ is an independent Rademacher vector. To upper bound the Rademacher complexity $\mathbf{E}_{x,y}\mathbf{E}_s \sup_{h \in V(r)} W^{-1} \sum_{j=1}^{W} s_j \mathrm{diff}_j(h)$, we define a metric between hypotheses in $V(r)$. For $h, h' \in H(r)$, we define $\theta^2(h, h') := \mathbf{E}\,W^{-1} \sum_{j=1}^{W} (\mathrm{diff}_j(h) - \mathrm{diff}_j(h'))^2$. Notice that $\theta^2(h, h') \leq 2r$ by the definition of $H(r)$. For every $h$, we define $X_h := W^{-1} \sum_{j=1}^{W} s_j \mathrm{diff}_j(h)$. Notice that $X_h - X_{h'}$ has sub-Gaussian increment with respect to $\theta(h, h')/\sqrt{W}$. By Dudley's inequality, we have

$$\mathbf{E}\sup_{h \in H(r)} X_h \leq \frac{1}{\sqrt{W}} \int_0^{\sqrt{2r}} \log \mathcal{N}(\epsilon, H(r), \theta) d\epsilon = O\left(\sqrt{\frac{dr\log r^{-1}}{W}}\right), \tag{4}$$

where $\mathcal{N}(\epsilon, H(r), \theta)$ is the covering number under metric $\theta$ and the last equality follows the VC dimension of $H$. This gives with probability $1 - \delta_r$,

$$\sup_{h \in H(r)} \left|W^{-1} \sum_{j=1}^{W} \mathrm{diff}_j(h) - W^{-1} \mathbf{E}\left[\sum_{j=1}^{W} \mathrm{diff}_j(h)\right]\right| \leq O\left(\sqrt{\frac{dr\log r^{-1} + 2r\log(1/\delta_r)}{W}} + \frac{3\log(1/\delta_r)}{W}\right).$$

By choosing $r_k = 2^{-k}, k = 1, 2, \ldots, \delta_{r_k} = \delta/2^{k+1}$, and union bound, we have with probability $1 - \delta$ for every $h \in H$,

$$W^{-1} \sum_{j=1}^{W} \mathrm{diff}_j(h) - W^{-1} \mathbf{E}\left[\sum_{j=1}^{W} \mathrm{diff}_j(h)\right] = O\left(\sqrt{\frac{dV(h)\log V(h)^{-1} + 2V(h)\log(1/\delta)}{W}} + \frac{3\log(1/(V(h)\delta))}{W}\right). \tag{5}$$

It remains to analyze $V(h)$. We have

$$\text{err}^{(t)}(h) - \text{err}^{(t)}(h^{(T)}) \geq \text{err}^{(t)}(h) - \text{err}^{(t)}(h^{(t)}) - 2\Delta W$$

$$= \underset{(x,y)\sim D^{(t)}}{\mathbf{E}} \mathbb{1}\{h(x) \neq h^{(t)}(x)\} \left(\mathbb{1}\{h(x) \neq y\} - \mathbb{1}\{h^{(t)}(x) \neq y\}\right) - 2\Delta W$$

$$\geq (1 - 2\eta) \underset{(x,y)\sim D^{(t)}}{\mathbf{E}} \mathbb{1}\{h(x) \neq h^{(t)}(x)\} - 2\Delta W$$

$$\geq (1 - 2\eta) \underset{(x,y)\sim D^{(t)}}{\mathbf{E}} \mathbb{1}\{h(x) \neq h^{(T)}(x)\} - 3\Delta W.$$

This implies that

$$\underset{D^{(t)}}{\mathbf{E}}[\text{diff}_j(h)] \geq (1 - 2\eta)\sigma_j^2(h) - 3W\Delta,$$

which implies that

$$V(h) \leq (1 - 2\eta)^{-1}W^{-1}\sum_{j=1}^{W} \underset{D^{(t)}}{\mathbf{E}} \text{diff}_j(h) + 3(1 - 2\eta)^{-1}W\Delta$$

$$\leq (1 - 2\eta)^{-1}\left(\text{err}^{(T)}(h) - \text{err}^{(T)}(h^T)\right) + 5(1 - 2\eta)^{-1}W\Delta. \tag{6}$$

By (3), (5) and (6), we get with probability at least $1 - \delta$,

$$\text{err}^{(T)}(\hat{h}^{(i)}) - \text{err}^{(T)}(h^T) \leq \tilde{O}(dW^{-1}(1 - 2\eta)^{-1} + \Delta W) = \tilde{O}(\sqrt{d\Delta/(1 - 2\eta)}).$$

$\square$

## B.2 Proof of Theorem 2.2

For convenience, we restate the theorem below.

**Theorem B.2** (Restatement of Theorem 2.2). *For every $T > 0$ and $\Delta \in (0, 1)$, there is a family of instances of learning halfspaces with $\eta$-RCN such that provided $(1 - 2\eta)^3 > d\Delta$, there is no learning algorithm $\mathcal{A}$ that can achieve error $\text{err}^{(T)}(\mathcal{A}) \leq \text{opt}_T + o(\sqrt{d\Delta/(1 - 2\eta)})$, with probability $1/2$ for every instance in the family.*

Our proof relies on the following simple fact, stated as the following lemma, which is a direct application of Theorem 2.2 in (Tsybakov, 2003).

**Lemma B.3.** *Let $S = (S_1, \ldots, S_d)$ be a random vector drawn uniformly from $\{\pm 1\}$. Consider $d$ coins, such that coin $i$ has probability $p_i = 1/2 + S_i\epsilon$ of landing on head when flipped. Let $\mathcal{A}$ be any algorithm that adaptively flips each coin, observes the outcomes and outputs some $\hat{S} \in \{\pm 1\}^d$ under the constraints that the number of flips for each coin is at most $1/(100\epsilon^2)$ must satisfy $\mathbf{Pr}(\|\hat{S} - S\|_1 \geq d/4) \geq 1/2$.*

*Proof of Theorem 2.2.* We construct the following family of sequences of distributions such that each distribution in the sequence corresponds to an instance of learning halfspaces in $\mathbb{R}^d$ with $\eta$-RCN. To do this, we first define the marginal distribution $D_x$ of the examples as follows. We first sample $i$ uniformly from $[d]$ and then sample $g \in (0, 1)$ uniformly and return $x = ge_i$, where $e_i$ is the $i^{\text{th}}$ standard basis vector. We next define the family of halfspaces $h_t^{(z)}$ for each $t$. For $z \in \{0, 1\}^d$, we define $h_t^{(z)}(x) := \text{sign}((\sum_{i=1}^{d} z_ie_i \cdot x) - (1 - \Delta td^{-1}(1 - 2\eta)^{-1}))$. That is to say $h_t^{(z)}(ge_i) = 1$ if and only if $z_i = 1$ and $g \geq 1 - \Delta td^{-1}(1 - 2\eta)^{-1}$.

Now we construct the sequence of distributions to learn. Let $m = \sqrt{d/(\Delta(1 - 2\eta))}/20$. For every $T > 0$, we first draw $z$ from $\{0, 1\}^d$ uniformly at random and construct distributions $D^{(i)}$ for $i = 1, 2, \ldots$, based on $z$. If $i \leq T - m$, let $h_0^{(z)}$ be the halfspace that realizes $D^{(i)}$ with marginal distribution $D_x$. If $i = T - m + t$, where $t \in [m]$, we let $h_t^{(z)}$ be the halfspace that realizes $D^{(i)}$ with marginal distribution $D_x$. We notice that for each $t$ and $z$,

$$\underset{x\sim D_x}{\mathbf{Pr}}(h_t^{(z)}(x) \neq h_{t+1}^{(z)}(x)) \leq \Delta(1 - 2\eta)^{-1}.$$

This implies that for every event $A \subseteq \mathbb{R}^d \times \{\pm 1\}$, we have

$$\left| \Pr_{x \sim D^{(i)}}(A) - \Pr_{x \sim D^{(i+1)}}(A) \right| = \left| \Pr_{(x,y) \sim D^{(i)}}(A, h_t^{(z)}(x) \neq h_{t+1}^{(z)}(x)) - \Pr_{(x,y) \sim D^{(i+1)}}(A, h_t^{(z)}(x) \neq h_{t+1}^{(z)}(x)) \right|$$
$$\leq (1 - 2\eta)\Delta(1 - 2\eta)^{-1} = \Delta,$$

which implies that $d_{TV}(D^{(i)}, D^{(i+1)}) \leq \Delta$.

For $i \in [d]$ and $j = T - m + t$, we define $X_j^{(i)} = \mathbb{1}\{x^{(j)} = ge_i, g > (1 - \Delta td^{-1}(1 - 2\eta)^{-1})\}$. We have

$$\mathbf{E} \sum_{t=1}^{m} \sum_{i=1}^{d} X_t^{(i)} \leq dm^2 \Delta d^{-1}(1 - 2\eta)^{-1} \leq d/(400(1 - 2\eta)^2),$$

where $c$ is a sufficiently small constant. By Markov's inequality, this implies that with probability at least $1/2$, there must be at least $d/2$ indices $i$ such that $\sum_{t=1}^{m} X_t^{(i)} \leq 1/(100(1 - 2\eta)^2)$. Given this happens, we know from Lemma B.3 that any algorithm $\mathcal{A}'$ fails to predict $z_i$ for $d/8$ of the indices with probability at least $1/2$.

We now show that any algorithm $\mathcal{A}$ that outputs a hypothesis $h$ with error $\mathrm{err}^{(T)}(h) \leq \eta + \sqrt{d\Delta/(1 - 2\eta)}/100$, gives an algorithm that correctly predicts $z_i$. Since

$$\mathrm{err}^{(T)}(h) = \eta + \Pr_{x \sim D_x}(h(x) \neq h_m^{(z)}(x))(1 - 2\eta),$$

we know that $\Pr_{x \sim D_x}(h(x) \neq h_m^{(z)}(x)) \leq (1 - 2\eta)^{-1}\sqrt{d\Delta/(1 - 2\eta)}/100$. This implies that for those $d/2$ indices $i$ such that $\sum_{t=1}^{m} X_t^{(i)} \leq 1/(100(1 - 2\eta)^2)$, there must be at least $3d/8$ of the indices $i$ such that

$$\Pr_{x \sim D_x}(h(x) = h_m^{(z)}(x) \mid x = ge_i, g > 1 - (1 - 2\eta)^{-1}\sqrt{d\Delta/(1 - 2\eta)}) > 1/2.$$

f Now we define $\mathcal{A}'$ by predicting $z_i$ as the majority of $h_m^{(z)}(x)$ given $ge_i, g > 1 - (1 - 2\eta)^{-1}\sqrt{d\Delta/(1 - 2\eta)})$. This implies that $\mathcal{A}'$ must succeed in predicting $3d/8$ of the $z_i$s, yielding a contradiction. $\square$

## B.3 Statistical Excess Error for Learning Drifting $\gamma$-Margin Halfspaces

**Theorem B.4.** *For every $T > 0$ and $\Delta \in (0, 1)$, there is a family of instances of learning $\gamma$-margin halfspaces with $\eta$-RCN such that provided $(1 - 2\eta)^2 > d\Delta$, there is no learning algorithm $\mathcal{A}$ that can achieve error $\mathrm{err}^{(T)}(\mathcal{A}) \leq \mathrm{opt}_T + o(\sqrt{\Delta}/\gamma)$, with probability $1/2$ for every instance in the family.*

*Proof of Theorem B.4.* For $\gamma \in (0, 1)$ small enough, we choose $d = \Theta(1/\gamma^2)$. Define $x^{(0)} = e_{d+1} - \sum_{i=1}^{d} e_i$ and $x^{(i)} = e_i - (e_{d+1}/2) \in \mathbb{R}^{d+1}$. For $z \in \{0, 1\}^d$, we define halfspace $h^{(z)}(x) = \mathrm{sign}(w^{(z)} \cdot x), w^{(z)} = (-e_{d+1} + \sum_{i=1}^{d} z_i e_i)$. We first show that any halfspace $h^{(z)}(x)$ has $\gamma$-margin with any marginal distribution over $X = \{x^{(i)}\}_{i=0}^{d}$. For $x^{(0)}$, we have $(-e_{d+1} + \sum_{i=1}^{d} z_i e_i) \cdot x^{(0)} = -\|z\|_1 - 1$, which implies

$$\frac{|w^{(z)} \cdot x^{(0)}|}{\|w^{(z)}\|_2 \|x^{(0)}\|_2} = \frac{\|z\|_1 + 1}{\|w^{(z)}\|_2 \|x^{(0)}\|_2} \geq \sqrt{\frac{\|z\|_1 + 1}{d}} = \Omega(\gamma).$$

For $x^{(i)}$, we have

$$\frac{|w^{(z)} \cdot x^{(i)}|}{\|w^{(z)}\|_2 \|x^{(0)}\|_2} \geq \frac{1/2}{\|w^{(z)}\|_2 \|x^{(0)}\|_2} \geq \frac{1/2}{\sqrt{d+1}} = \Omega(1/\gamma).$$

We next design the family of sequences of drifted distributions $D^{(1)}, \ldots, D^{(T)}$. For any $z \in \{0, 1\}^d$, we fix the target halfspace to be $h^{(z)}$. Choose $m = \sqrt{d/\Delta}(1 - 2\eta)^{-1}$. For $t \leq T - m$, we define the marginal distribution $(D_t)_x$ to be the Dirac function at point $x^{(0)}$. For $t = T - m + j, j \in [m]$, choose $(D^{(t)})_x = (1 - p_t)\delta(x^{(0)}) + \sum_{i=1}^{d}(p_t/d)\delta(x^{(i)})$, where $p_t = \Delta j$. We notice that $d_{TV}(D_t, D_{t+1}) \leq \Delta$.

Let $z \sim \{0,1\}^d$ be chosen uniformly at random, we will show that no algorithm in expectation can achieve $O(\sqrt{d\Delta})$ error. By the choice of $z$, we know that $h^{(z)}(x^{(i)}) = 1$ with probability $1/2$ for every $i \in [d]$. By the design of $D_t$, we know that $D_x^{(t)}(x^{(i)}) \leq D_x^{(T)}(x^{(i)}) = \sqrt{\Delta/d}(1-2\eta)^{-1}$. Let $X_t^{(i)} = \mathbb{1}\{x_t = x^{(i)}\}$, $x_t \sim D_x^{(t)}$ This implies

$$\mathbf{E} \sum_{t=1}^{T} \sum_{i=1}^{d} X_t^{(i)} = \mathbf{E} \sum_{t=T-m}^{T} \sum_{i=1}^{d} X_t^{(i)} \leq dm\sqrt{\Delta/d}(1-2\eta)^{-1} \leq cd/(1-2\eta)^2$$

for some $c < 1/100$. This implies that there must be $d/2$, $i \in [d]$ such that $\sum_{t=1}^{T} X_t^{(i)} \leq c/(1-2\eta)^2$. Since the distribution of the label of $x^{(i)}$ is independent of the marginal distribution, by Lemma B.3, we know that for these $x^{(i)}$, we cannot tell if the ground truth label is $+1$ or $-1$, which implies that the prediction error is at least $\Omega(\sqrt{d\Delta})$. $\square$

# C Omitted Proofs from Section 3

## C.1 Proof of Fact 3.1

We restate the fact below.

*Fact* C.1 (Restatement of Fact 3.1). Let $t > 0$ be any time step and let $S = \{(x^{(i)}, y^{(i)})\}_{i=1}^{m}$ be such that $(x^{(i)}, y^{(i)}) \sim D^{(t+i)}$. For $T \in [m]$ and $w^* \in \mathbb{S}^{d-1}$, Algorithm 2 satisfies

$$\frac{1}{T} \sum_{i=1}^{T} \mathop{\mathbf{E}}_{(x,y)\sim D^{(t+i)}} g(w^{(i)}; x, y) \cdot (w^{(i)} - w^*) \leq R(T, \mu, \gamma),$$

where $R(T, \mu, \gamma) = 1/(2T\mu) + 2\mu/\gamma^2 - \sum_{i=1}^{T} \xi_i/T$, $\xi_i := \left(g(w^{(i)}; x, y) - \mathbf{E}_{(x,y)\sim D^{(t+i)}} g(w^{(i)}; x, y)\right) \cdot (w^{(i)} - w^*)$.

*Proof of Fact 3.1.* By the nonexpansivity of the projection operator, we have

$$\left\|w^{(i+1)} - w^*\right\|^2 \leq \left\|w^{(i)} - w^* - \mu g(w^{(i)}; x, y)\right\|^2$$

$$\leq \left\|w^{(i)} - w^*\right\|^2 + \mu^2 \left\|g(w^{(i)}; x, y)\right\|^2 - 2\mu g(w^{(i)}; x, y) \cdot (w^{(i)} - w^*)$$

$$= \left\|w^{(i)} - w^*\right\|^2 + \mu^2 \left\|g(w^{(i)}; x, y)\right\|^2 - 2\mu \left(g(w^{(i)}; x, y) - \mathop{\mathbf{E}}_{(x,y)\sim D^{(t)}} g(w^{(i)}; x, y)\right) \cdot (w^{(i)} - w^*)$$

$$- 2\mu \mathop{\mathbf{E}}_{(x,y)\sim D^{(t+i)}} g(w^{(i)}; x, y) \cdot (w^{(i)} - w^*)$$

$$= \left\|w^{(i)} - w^*\right\|^2 + \mu^2 \left\|g(w^{(i)}; x, y)\right\|^2 - 2\mu\xi_t - 2\mu \mathop{\mathbf{E}}_{(x,y)\sim D^{(t+i)}} g(w^{(i)}; x, y) \cdot (w^{(i)} - w^*). \quad (7)$$

Rearranging and taking the average of (7) over $[T]$, we have

$$\frac{1}{T} \sum_{i=1}^{T} \mathop{\mathbf{E}}_{(x,y)\sim D^{(t+i)}} g(w^{(i)}; x, y) \cdot (w^{(i)} - w^*) \leq \frac{\left\|w^{(0)} - w^*\right\|^2}{2T\mu} + \frac{\mu}{2T} \sum_{i=1}^{T} \left\|g(w^{(i)}; x, y)\right\|^2 - \frac{1}{T} \sum_{i=1}^{T} \xi_i$$

$$\leq \frac{1}{2T\mu} + \frac{2\mu}{\gamma^2} - \frac{1}{T} \sum_{i=1}^{T} \xi_i,$$

where we used the fact that $\left\|g_i(w^{(i)}; x, y)\right\|^2 \leq 4/\gamma^2$. $\square$

## C.2 Omitted Proofs for Lemma 3.2

We restate the lemma below.

**Lemma C.2** (Restatement of Lemma 3.2). *Let $i, m \in \mathbb{N}$ and let $w^* \in \mathbb{S}^{d-1}$ be a vector that realizes $D^{(i+m)}$. There exists a function $F_i : \mathbb{S}^{d-1} \times \mathbb{S}^{d-1} \times \mathbb{S}^{d-1} \to \mathbb{R}$ such that for any $x \in \mathbb{S}^{d-1}$ and for any $w \in \mathbb{S}^{d-1}$,*

$$\mathop{\mathbf{E}}_{y\sim D^{(i)}|x} g(w; x, y) \cdot (w - w^*) \geq 2(\mathrm{err}^{(i)}(w, x) - \eta) - F_i(w^*, w, x).$$

*Additionally, $|F_i| \leq 1/\gamma$, and for every $w \in \mathbb{S}^{d-1}$, $\mathbf{E}_{x\sim D_x^{(i)}} F_i(w^*, w, x) \leq O(\Delta m/\gamma)$.*

*Proof of Lemma 3.2.* The proof of Lemma 3.2 is divided into two cases based on $|w \cdot x|$.

**Case 1: Large Margin** In the first case, $|w \cdot x| \geq \gamma$ and we establish the following claim.

*Claim* C.3. Let $i, m \in \mathbb{N}$ and let $(w^*)^{(i)} \in \mathbb{S}^{d-1}$ be a vector that realizes $D^{(i)}$. Denote by $f_i(x, w^*, x) = ((1 - 2\eta_{m+i}(x)) - (1 - 2\eta_i(x))\phi(w^* \cdot x, (w^*)^{(i)} \cdot x)) |w^* \cdot x| / |w \cdot x|$. Then, for every $x \in \mathbb{S}^{d-1}$ such that $|w \cdot x| \geq \gamma$, $\mathbf{E}_{y \sim D^{(i)}|x} g(w; x, y) \cdot (w - w^*) \geq 2(\mathrm{err}^{(i)}(w, x) - \eta) - f_i(w^*, w, x)$.

We defer the proof of Claim C.3 to Appendix C.2.1. Since $|w \cdot x| \geq \gamma$, we know that $|f_i(w^*, w, x)| \leq 1/\gamma$. We next upper bound the expectation of $f_i(w^*, w, x)$. Formally, we have the following claim, whose proof is deferred to Appendix C.2.2.

*Claim* C.4. For every $w \in \mathbb{S}^{d-1}$, it holds $\mathbf{E}_{(x,y) \sim D^{(i)}} f_i(w^*, w, x) \mathbb{1}\{|w \cdot x| \geq \gamma\} = O(m\Delta/\gamma)$.

To prove Claim C.4, we establish the following two claims that will be used frequently. We defer the proofs of Claim 3.3 and Claim 3.4 to Appendix C.2.3 and Appendix C.2.4.

**Case 2: Small Margin** We next consider the case where $|w \cdot x| < \gamma$. We will discuss two cases. In the first case, we have $\mathrm{sign}(w \cdot x) = \mathrm{sign}((w^*)^{(i)} \cdot x)$, which implies that $\mathrm{err}^{(i)}(w, x) - \eta = \eta_i(x) - \eta \leq 0$. In this case, we establish the following claim.

*Claim* C.5. Let $i, m \in \mathbb{N}$ and let $(w^*)^{(i)} \in \mathbb{S}^{d-1}$ be a vector that realizes $D^{(i)}$. Denote by $G_i(w^*, w, x) = ((1 - 2\eta_{m+i}(x)) - (1 - 2\eta_i(x))\phi((w^*)^{(i)} \cdot x, w^* \cdot x)) |w^* \cdot x| / \gamma$, then for every $x \in \mathbb{S}^{d-1}$ such that $|w \cdot x| < \gamma$ and $\mathrm{sign}(w \cdot x) = \mathrm{sign}((w^*)^{(i)} \cdot x)$, we have $\mathbf{E}_{y \sim D^{(i)}|x} g(w; x, y) \cdot (w - w^*) \geq 2(\mathrm{err}^{(i)}(w, x) - \eta) - G_i(w^*, w, x)$.

Notice that $|G_i(w^*, w, x)| \leq 1/\gamma$ always holds. We next upper bound the expectation of $G_i(w^*, w, x)$ with the following claim.

*Claim* C.6. For every $w \in \mathbb{S}^{d-1}$, it holds $\mathbf{E}_{(x,y) \sim D^{(i)}} G_i(w^*, w, x) \mathbb{1}\{|w \cdot x| < \gamma, \mathrm{sign}(w \cdot x) = \mathrm{sign}((w^*)^{(i)} \cdot x)\} = O(m\Delta/\gamma)$.

We defer the proof of Claim C.5 and Claim C.6 to Appendix C.2.5 and Appendix C.2.6.

Next, we consider the case, where $\mathrm{sign}(w \cdot x) \neq \mathrm{sign}((w^*)^{(i)} \cdot x)$, which implies that $\mathrm{err}^{(i)}(w, x) - \eta = 1 - \eta_i(x) - \eta \geq 0$. We give the following claim in this case.

*Claim* C.7. Let $i, m \in \mathbb{N}$ and let $(w^*)^{(i)} \in \mathbb{S}^{d-1}$ be a vector that realizes $D^{(i)}$ and $w^* \in \mathbb{S}^{d-1}$ be a vector that realizes $D^{(i+m)}$. Denote by

$$V_i(w^*, x) = (1 - \eta - \eta_i(x)) \mathbb{1}\{\phi(w^* \cdot x, (w^*)^{(i)} \cdot x) = 1, |w^* \cdot x| < \gamma\}(\gamma - |w^* \cdot x|)/\gamma,$$
$$U_i(w^*, x) = (1 - \eta - \eta_i(x)) \mathbb{1}\{\phi(w^* \cdot x, (w^*)^{(i)} \cdot x) = -1\}(|w^* \cdot x| + \gamma)/\gamma.$$

Then for every $x \in \mathbb{S}^{d-1}$ such that $|w \cdot x| < \gamma$ and $\mathrm{sign}(w \cdot x) \neq \mathrm{sign}((w^*)^{(i)} \cdot x)$, we have $\mathbf{E}_{y \sim D^{(i)}|x} g(w; x, y) \cdot (w - w^*) \geq 2(\mathrm{err}^{(i)}(w, x) - \eta) - V_i(w^*, x) - U_i(w^*, x)$.

Since $|V_i|, |U_i| = O(1/\gamma)$ always hold, we thus upper bound the expectation of $V_i$ and $U_i$. We make the following claim.

*Claim* C.8. Let $i, m \in \mathbb{N}$ and let $(w^*)^{(i)} \in \mathbb{S}^{d-1}$ be a vector that realizes $D^{(i)}$ and $w^* \in \mathbb{S}^{d-1}$ be a vector that realizes $D^{(i+m)}$. Then

$$\mathbf{E}_{(x,y) \sim D^{(i)}} V_i(w^*, x) \mathbb{1}\{|w \cdot x| < \gamma, \mathrm{sign}(w \cdot x) \neq \mathrm{sign}((w^*)^{(i)} \cdot x)\} = O(m\Delta)$$
$$\mathbf{E}_{(x,y) \sim D^{(i)}} U_i(w^*, x) \mathbb{1}\{|w \cdot x| < \gamma, \mathrm{sign}(w \cdot x) \neq \mathrm{sign}((w^*)^{(i)} \cdot x)\} = O(m\Delta)$$

We defer the proof of Claim C.7 and Claim C.8 to Appendix C.2.7 and Appendix C.2.8 and conclude the proof of Lemma 3.2 using the claims we have established so far. Define

$$F_i(w^*, w, x) := f_i(w^*, w, x) \mathbb{1}\{|w \cdot x| \geq \gamma\} + G_i(w^*, w, x) \mathbb{1}\{|w \cdot x| < \gamma, \mathrm{sign}(w \cdot x) = \mathrm{sign}((w^*)^{(i)} \cdot x)\}$$
$$+ (U_i(w^*, x) + V_i(w^*, x)) \mathbb{1}\{|w \cdot x| < \gamma, \mathrm{sign}(w \cdot x) \neq \mathrm{sign}((w^*)^{(i)} \cdot x)\}.$$

By Claim C.3, Claim C.5, and Claim C.7, we know that for every $w \in \mathbb{S}^{d-1}$ and for every $x \in \mathbb{S}^{d-1}$, we have

$$\mathop{\mathbf{E}}_{y \sim D^{(i)}|x} g(w; x, y) \cdot (w - w^*) \geq 2(\mathrm{err}^{(i)}(w, x) - \eta) - F_i(w^*, w, x).$$

Since $|f_i|, |G_i|, |U_i|, |V_i| = O(1/\gamma)$, we know that $|F_i| = O(1/\gamma)$. Furthermore, by Claim C.4, Claim C.6, and Claim C.8, we have

$$
\begin{aligned}
\mathop{\mathbf{E}}_{x \sim D_x^{(i)}} F_i(w^*, w, x) &= \mathop{\mathbf{E}}_{x \sim D_x^{(i)}} f_i(w^*, w, x) \mathbb{1}\{|w \cdot x| \geq \gamma\} \\
&\quad + \mathop{\mathbf{E}}_{x \sim D_x^{(i)}} G_i(w^*, w, x) \mathbb{1}\{|w \cdot x| < \gamma, \mathrm{sign}(w \cdot x) = \mathrm{sign}((w^*)^{(i)} \cdot x)\} \\
&\quad + \mathop{\mathbf{E}}_{x \sim D_x^{(i)}} (U_i(w^*, x) + V_i(w^*, x)) \mathbb{1}\{|w \cdot x| < \gamma, \mathrm{sign}(w \cdot x) \neq \mathrm{sign}((w^*)^{(i)} \cdot x)\} \\
&= O(m\Delta/\gamma).
\end{aligned}
$$

$\square$

### C.2.1  PROOF OF CLAIM C.3

*Proof of Claim C.3.* We start by bounding $\mathbf{E}_{y \sim D^{(i)}|x} g(w; x, y) \cdot w$.

$$
\begin{aligned}
\mathop{\mathbf{E}}_{y \sim D^{(i)}|x} g(w; x, y) \cdot w &= (((1 - 2\eta)\mathrm{sign}(w \cdot x) - \mathbf{E}\, y(x))x / \max\{|w \cdot x|, \gamma\}) \cdot w \\
&= \mathrm{sign}(w \cdot x) \left( (1 - 2\eta)\mathrm{sign}(w \cdot x) - (1 - 2\eta_i(x))\mathrm{sign}((w^*)^{(i)} \cdot x) \right) \\
&= (1 - 2\eta) - (1 - 2\eta_i(x))\mathrm{sign}\left( (w^*)^{(i)} \cdot x \right) \mathrm{sign}(w \cdot x) \\
&= (1 - 2\eta) - (1 - 2\eta_i(x))\phi\left( (w^*)^{(i)} \cdot x, w \cdot x \right).
\end{aligned}
$$

We consider two cases. If $\mathrm{sign}\left((w^*)^{(i)} \cdot x\right) = \mathrm{sign}(w \cdot x)$, which implies $\phi\left((w^*)^{(i)} \cdot x, w \cdot x\right) = 1$, then we have

$$\mathop{\mathbf{E}}_{y \sim D^{(i)}|x} g(w; x, y) \cdot w = 2(\eta_i(x) - \eta) = 2(\mathrm{err}^{(i)}(w, x) - \eta).$$

If $\mathrm{sign}\left((w^*)^{(i)} \cdot x\right) \neq \mathrm{sign}(w \cdot x)$, which implies $\phi\left((w^*)^{(i)} \cdot x, w \cdot x\right) = -1$ then we have

$$\mathop{\mathbf{E}}_{y \sim D^{(i)}|x} g(w; x, y) \cdot w = 2(1 - \eta_i(x) - \eta) = 2(\mathrm{err}^{(i)}(w, x) - \eta).$$

Thus, it always holds that

$$\mathop{\mathbf{E}}_{y \sim D^{(i)}|x} g(w; x, y) \cdot w = 2(\mathrm{err}^{(i)}(w, x) - \eta). \tag{8}$$

It remains to analyze the term with respect to $w^*$. We have

$$
\begin{aligned}
\mathop{\mathbf{E}}_{y \sim D^{(i)}|x} g(w; x, y) \cdot w^* &= (((1 - 2\eta)\mathrm{sign}(w \cdot x) - \mathbf{E}\, y(x))x / |w \cdot x|) \cdot w^* \\
&= \mathrm{sign}(w^* \cdot x) \left( (1 - 2\eta)\mathrm{sign}(w \cdot x) - (1 - 2\eta_i(x))\mathrm{sign}((w^*)^{(i)} \cdot x) \right) |w^* \cdot x| / |w \cdot x| \\
&= ((1 - 2\eta)\phi(w \cdot x, w^* \cdot x) - (1 - 2\eta_{m+i}(x))) |w^* \cdot x| / |w \cdot x| \\
&\quad + \left( (1 - 2\eta_{m+i}(x)) - (1 - 2\eta_i(x))\phi(w^* \cdot x, (w^*)^{(i)} \cdot x) \right) |w^* \cdot x| / |w \cdot x| \\
&\leq \left( (1 - 2\eta_{m+i}(x)) - (1 - 2\eta_i(x))\phi(w^* \cdot x, (w^*)^{(i)} \cdot x) \right) |w^* \cdot x| / |w \cdot x|. \tag{9}
\end{aligned}
$$

Here, the inequality holds because $(1 - 2\eta)\phi(w \cdot x, w^* \cdot x) \leq (1 - 2\eta) \leq (1 - 2\eta_i(x))$. Combine (8) and (9), we have $\mathbf{E}_{y \sim D^{(i)}|x} g(w; x, y) \cdot (w - w^*) \geq 2(\mathrm{err}^{(i)}(w, x) - \eta) - f_i(w^*, w, x)$ since $|w \cdot x| \geq \gamma$. $\square$

### C.2.2   PROOF OF CLAIM C.4

*Proof of Claim C.4.* By taking the expectation over $x \sim D_x^{(i)}$, we have

$$\mathop{\mathbf{E}}_{x \sim D^{(i)x}} f_i(w^*, w, x) \mathbb{1}\{|w \cdot x| \geq \gamma\}$$

$$= \mathop{\mathbf{E}}_{x \sim D_x^{(i)}} \left( (1 - 2\eta_{m+i}(x)) - (1 - 2\eta_i(x))\phi(w^* \cdot x, (w^*)^{(i)} \cdot x) \right) |w^* \cdot x| / |w \cdot x| \mathbb{1}\{|w \cdot x| \geq \gamma\}$$

$$= \mathop{\mathbf{E}}_{x \sim D_x^{(i)}} ((1 - 2\eta_{m+i}(x)) + (1 - 2\eta_i(x))) |w^* \cdot x| / |w \cdot x| \mathbb{1}\{\phi(w^* \cdot x, (w^*)^{(i)} \cdot x) = -1, |w \cdot x| \geq \gamma\}$$

$$+ \mathop{\mathbf{E}}_{x \sim D_x^{(i)}} (2\eta_i(x) - 2\eta_{m+i}(x)) |w^* \cdot x| / |w \cdot x| \mathbb{1}\{\phi(w^* \cdot x, (w^*)^{(i)} \cdot x) = 1, |w \cdot x| \geq \gamma\}$$

$$\leq 2 \mathop{\mathbf{Pr}}_{x \sim D_z^{(i)}} (\phi(w^* \cdot x, (w^*)^{(i)} \cdot x) = -1)(1 - 2\eta)/\gamma$$

$$+ 2 \mathop{\mathbf{E}}_{x \sim D_x^{(i)}} (\eta_i(x) - \eta_{m+i}(x)) \mathbb{1}\{\phi(w^* \cdot x, (w^*)^{(i)} \cdot x) = 1, \eta_i(x) \geq \eta_{m+i}(x)\}/\gamma \tag{10}$$

By Claim 3.3 and Claim 3.4, we know that $\mathbf{Pr}_{x \sim D_x^{(i)}}(\phi(w^* \cdot x, (w^*)^{(i)} \cdot x) = -1) \leq O(\Delta m/(1 - 2\eta))$ and $\mathbf{E}_{x \sim D_x^{(i)}} (\eta_i(x) - \eta_{m+i}(x)) \mathbb{1}\{\phi(w^* \cdot x, (w^*)^{(i)} \cdot x) = 1, \eta_i(x) \geq \eta_{m+i}(x)\} \leq O(\Delta m)$. Thus, by(10), we have

$$\mathop{\mathbf{E}}_{(x,y) \sim D^{(i)}} f_i(w^*, w, x) \mathbb{1}\{|w \cdot x| \geq \gamma\} \leq O(m\Delta/\gamma).$$

$\square$

### C.2.3   PROOF OF CLAIM 3.3

*Proof of Claim 3.3.* Suppose that $\mathbf{Pr}_{x \sim D_x^{(i)}}(\phi(w^* \cdot x, (w^*)^{(i)} \cdot x) = -1) \geq (1 + \eta)\Delta m/(1 - 2\eta)$, then we have

$$\left( \mathop{\mathbf{Pr}}_{(x,y) \sim D^{(i)}} - \mathop{\mathbf{Pr}}_{(x,y) \sim D^{(m+i)}} \right) \left( \phi(w^* \cdot x, (w^*)^{(i)} \cdot x) = -1, \mathrm{sign}(w^* \cdot x) \neq y \right)$$

$$\geq (1 - \eta) \mathop{\mathbf{Pr}}_{x \sim D_x^{(i)}} (\phi(w^* \cdot x, (w^*)^{(i)} \cdot x) = -1) - \eta \mathop{\mathbf{Pr}}_{x \sim D_x^{(m+1)}} (\phi(w^* \cdot x, (w^*)^{(i)} \cdot x) = -1)$$

$$> (1 - \eta)(1 + \eta)\Delta m/(1 - 2\eta) - \eta(1 + \eta)\Delta m/(1 - 2\eta) - \eta\Delta m = \Delta m,$$

which contradicts with the fact that $d_{\mathrm{TV}}(D^{(i)}, D^{(j)}) \leq m\Delta$. This implies $\mathbf{Pr}_{x \sim D_x^{(i)}}(\phi(w^* \cdot x, (w^*)^{(i)} \cdot x) = -1) \leq O(\Delta m/(1 - 2\eta))$. $\square$

### C.2.4   PROOF OF CLAIM 3.4

*Proof of Claim 3.4.* For convenience, we denote by $A$ the set of $x$ such that $\phi(w^* \cdot x, (w^*)^{(i)} \cdot x) = 1, \eta_i(x) \geq \eta_{m+i}(x)$. We have

$$\mathop{\mathbf{E}}_{x \sim D_x^{(i)}} ((\eta_i(x) - \eta_{m+i}(x)) \mathbb{1}\{A\}$$

$$= \mathop{\mathbf{E}}_{x \sim D_x^{(i)}} \eta_i(x)\mathbb{1}\{A\} - \mathop{\mathbf{E}}_{x \sim D_x^{(m+i)}} \eta_{m+i}(x)\mathbb{1}\{A\} + \mathop{\mathbf{E}}_{x \sim D_x^{(m+i)}} \eta_{m+i}(x)\mathbb{1}\{A\} - \mathop{\mathbf{E}}_{x \sim D_i^{(i)}} \eta_{m+i}(x)\mathbb{1}\{A\}$$

$$= \mathop{\mathbf{Pr}}_{x \sim D^{(i)}} (x \in A, \mathrm{sign}(w^* \cdot x) \neq y) - \mathop{\mathbf{Pr}}_{x \sim D^{(i+m)}} (x \in A, \mathrm{sign}(w^* \cdot x) \neq y) + \int_x \eta_{m+i}(x)\mathbb{1}\{A\}(D^{(m+i)}(x) - D^{(i)}(x))$$

$$\leq m\Delta + m\Delta \leq 2m\Delta.$$

$\square$

### C.2.5    PROOF OF CLAIM C.5

*Proof of Claim C.5.* we have

$$\mathop{\mathbf{E}}_{y \sim D^{(i)}|x} g(w; x, y) \cdot (w - w^*)$$

$$= (((1 - 2\eta)\text{sign}(w \cdot x) - \mathbf{E}\, y(x))x / \max\{|w \cdot x|, \gamma\}) \cdot (w - w^*)$$

$$= \Big(((1 - 2\eta)\text{sign}(w \cdot x) - (1 - 2\eta_i(x))\text{sign}((w^*)^{(i)} \cdot x)\Big)(w - w^*) \cdot x / \gamma$$

$$= 2(\eta_i(x) - \eta)|w \cdot x|/\gamma - \Big(((1 - 2\eta)\text{sign}(w \cdot x) - (1 - 2\eta_i(x))\text{sign}((w^*)^{(i)} \cdot x)\Big)w^* \cdot x / \gamma$$

$$\geq 2(\text{err}^{(i)}(w, x) - \eta) - \Big(((1 - 2\eta)\text{sign}(w \cdot x) - (1 - 2\eta_i(x))\text{sign}((w^*)^{(i)} \cdot x)\Big)w^* \cdot x / \gamma.$$

We notice that

$$\Big(((1 - 2\eta)\text{sign}(w \cdot x) - (1 - 2\eta_i(x))\text{sign}((w^*)^{(i)} \cdot x)\Big)w^* \cdot x / \gamma$$

$$= \text{sign}(w^* \cdot x)\Big((1 - 2\eta)\text{sign}(w \cdot x) - (1 - 2\eta_i(x))\text{sign}((w^*)^{(i)} \cdot x)\Big)|w^* \cdot x|/\gamma$$

$$= ((1 - 2\eta)\phi(w \cdot x, w^* \cdot x) - (1 - 2\eta_{m+i}(x)))|w^* \cdot x|/\gamma$$

$$+ \Big((1 - 2\eta_{m+i}(x)) - (1 - 2\eta_i(x))\phi((w^*)^{(i)} \cdot x, w^* \cdot x)\Big)|w^* \cdot x|/\gamma$$

$$\leq \Big((1 - 2\eta_{m+i}(x)) - (1 - 2\eta_i(x))\phi((w^*)^{(i)} \cdot x, w^* \cdot x)\Big)|w^* \cdot x|/\gamma,$$

where the inequality follows $((1 - 2\eta)\phi(w \cdot x, w^* \cdot x) - (1 - 2\eta_{m+i}(x)))|w^* \cdot x|/\gamma \leq 0$. This implies that when $|w \cdot x| < \gamma$ and $\text{sign}(w \cdot x) = \text{sign}((w^*)^{(i)} \cdot x)$, we have

$$\mathop{\mathbf{E}}_{y \sim D^{(i)}|x} g(w; x, y) \cdot (w - w^*) \geq 2(\text{err}^{(i)}(w, x) - \eta) - G_i(w^*, w, x).$$

$\square$

### C.2.6    PROOF OF CLAIM C.6

*Proof of Claim C.6.* To simplify the notation, we denote by $A$ the event where $|w \cdot x| < \gamma$ and $\text{sign}(w^t \cdot x) = \text{sign}((w^*)^{(i)} \cdot x)$. We have

$$\mathop{\mathbf{E}}_{(x,y) \sim D^{(i)}} G_i(w^*, w, x)\mathbb{1}\{A\}$$

$$= \mathop{\mathbf{E}}_{(x,y) \sim D^{(i)}} \Big((1 - 2\eta_{m+i}(x)) - (1 - 2\eta_i(x))\phi((w^*)^{(i)} \cdot x, w^* \cdot x)\Big)|w^* \cdot x|/\gamma\mathbb{1}\{A\}$$

$$= \mathop{\mathbf{E}}_{x \sim D^{(i)}} ((1 - 2\eta_{m+i}(x)) + (1 - 2\eta_i(x)))|w^* \cdot x'|/\gamma\mathbb{1}\{\phi(w^* \cdot x, (w^*)^{(i)} \cdot x) = -1, A\}$$

$$+ \mathop{\mathbf{E}}_{x \sim D^{(i)}} ((2\eta_i(x) - 2\eta_{m+i}(x))|w^* \cdot x'|/\gamma\mathbb{1}\{\phi(w^* \cdot x, (w^*)^{(i)} \cdot x) = 1, A\}$$

$$\leq 2\mathop{\mathbf{Pr}}_{x \sim D^{(i)}} (\phi(w^* \cdot x, (w^*)^{(i)} \cdot x) = -1)(1 - 2\eta)/\gamma$$

$$+ 2\mathop{\mathbf{E}}_{x \sim D^{(i)}} ((\eta_i(x) - \eta_{m+i}(x))\mathbb{1}\{\phi(w^* \cdot x, (w^*)^{(i)} \cdot x) = 1, \eta_i(x) \geq \eta_{m+i}(x))/\gamma\} \leq O(m\Delta/\gamma).$$

Here, the last inequality follows Claim 3.3 and Claim 3.4.                    $\square$

### C.2.7   PROOF OF CLAIM C.7

*Proof of Claim C.7.* We have

$$\mathop{\mathbf{E}}_{y \sim D^{(i)}} g(w; x, y) \cdot (w - w^*)$$

$$= (((1 - 2\eta)\mathrm{sign}(w \cdot x) - \mathbf{E}\, y(x))x / \max\{|w \cdot x|, \gamma\}) \cdot (w - w^*)$$

$$= \Big(((1 - 2\eta)\mathrm{sign}(w \cdot x) - (1 - 2\eta_i(x))\mathrm{sign}((w^*)^{(i)} \cdot x)\Big) (w - w^*) \cdot x / \gamma$$

$$= 2(1 - \eta_i - \eta)(|w \cdot x| / \gamma) - \mathrm{sign}(w^* \cdot x) \Big((1 - 2\eta)\mathrm{sign}(w \cdot x) - (1 - 2\eta_i(x))\mathrm{sign}((w^*)^{(i)} \cdot x)\Big) |w^* \cdot x| / \gamma$$

$$\geq -\mathrm{sign}(w^* \cdot x) \Big((1 - 2\eta)\mathrm{sign}(w \cdot x) - (1 - 2\eta_i(x))\mathrm{sign}((w^*)^{(i)} \cdot x)\Big) |w^* \cdot x| / \gamma$$

$$= 2\phi(w^* \cdot x, (w^*)^{(i)} \cdot x)(1 - \eta - \eta_i(x)) |w^* \cdot x| / \gamma$$

Denote by $K_i(w^*, x) := \phi(w^* \cdot x, (w^*)^{(i)} \cdot x)(1 - \eta - \eta_i(x)) |w^* \cdot x| / \gamma$ and we will show that

$$K_i(w^*, x) \geq (\mathrm{err}^{(i)}(w, x) - \eta) - V_i(w^*, x)/2 - U_i(w^*, x)/2.$$

To see this, we expand $K_i(w^*, x)$ as follows

$$K_i(w^*, x) = (1 - \eta - \eta_i(x))\mathbb{1}\{\phi(w^* \cdot x, (w^*)^{(i)} \cdot x) = 1, |w^* \cdot x| \geq \gamma\} |w^* \cdot x| / \gamma$$
$$+ (1 - \eta - \eta_i(x))\mathbb{1}\{\phi(w^* \cdot x, (w^*)^{(i)} \cdot x) = 1, |w^* \cdot x| < \gamma\} |w^* \cdot x| / \gamma$$
$$- (1 - \eta - \eta_i(x))\mathbb{1}\{\phi(w^* \cdot x, (w^*)^{(i)} \cdot x) = -1\} |w^* \cdot x| / \gamma.$$

We analyze the three terms separately. First, when $|w^* \cdot x| \geq \gamma$, we have

$$\mathbb{1}\{\phi(w^* \cdot x, (w^*)^{(i)} \cdot x) = 1, |w^* \cdot x| \geq \gamma\} |w^* \cdot x| / \gamma \geq \mathbb{1}\{\phi(w^* \cdot x, (w^*)^{(i)} \cdot x) = 1, |w^* \cdot x| \geq \gamma\}. \tag{11}$$

Next, we decompose

$$\mathbb{1}\{\phi(w^* \cdot x, (w^*)^{(i)} \cdot x) = 1, |w^* \cdot x| < \gamma\} |w^* \cdot x| / \gamma = \mathbb{1}\{\phi(w^* \cdot x, (w^*)^{(i)} \cdot x) = 1, |w^* \cdot x| < \gamma\}(1 - (\gamma - |w^* \cdot x|)/\gamma). \tag{12}$$

Since

$$\mathbb{1}\{\phi(w^* \cdot x, (w^*)^{(i)} \cdot x) = 1, |w^* \cdot x| < \gamma\} + \mathbb{1}\{\phi(w^* \cdot x, (w^*)^{(i)} \cdot x) = 1, |w^* \cdot x| \geq \gamma\} + \mathbb{1}\{\phi(w^* \cdot x, (w^*)^{(i)} \cdot x) = -1\} = 1,$$

by (11) and (12), we have

$$K_i(w^*, x) \geq (1 - \eta - \eta_i(x))$$
$$- (1 - \eta - \eta_i(x))\mathbb{1}\{\phi(w^* \cdot x, (w^*)^{(i)} \cdot x) = 1, |w^* \cdot x| < \gamma\}(\gamma - |w^* \cdot x|)/\gamma$$
$$- (1 - \eta - \eta_i(x))\mathbb{1}\{\phi(w^* \cdot x, (w^*)^{(i)} \cdot x) = -1\}(|w^* \cdot x| + \gamma)/\gamma$$
$$= (\mathrm{err}^{(i)}(w, x)) - V_i(w^*, x) - U_i(w^*, x).$$

$\square$

### C.2.8   PROOF OF CLAIM C.8

*Proof of Claim C.8.* To simplify the notation, we denote by $B$ the event where $|w \cdot x| < \gamma$ and $\mathrm{sign}(w \cdot x) \neq \mathrm{sign}((w^*)^{(i)} \cdot x)$. We have

$$\mathop{\mathbf{E}}_{(x,y) \sim D^{(i)}} V_i \mathbb{1}\{B\} = \mathop{\mathbf{E}}_{(x,y) \sim D^{(i)}} (1 - \eta - \eta_i(x))\mathbb{1}\{\phi(w^* \cdot x, (w^*)^{(i)} \cdot x) = 1, |w^* \cdot x| < \gamma, B\}(\gamma - |w^* \cdot x|)/\gamma$$

$$\leq \mathop{\mathbf{E}}_{(x,y) \sim D^{(i)}} (1 - \eta - \eta_i(x))\mathbb{1}\{\phi(w^* \cdot x, (w^*)^{(i)} \cdot x) = 1, |w^* \cdot x| < \gamma, B\}$$

$$\leq \mathop{\mathbf{Pr}}_{(x,y) \sim D^{(i)}} (|w^* \cdot x| < \gamma) \leq O(m\Delta).$$

$$\mathop{\mathbf{E}}_{(x,y)\sim D^{(i)}} U_i \mathbb{1}\{B\} = (1 - \eta - \eta_i(x))\mathbb{1}\{\phi(w^* \cdot x, (w^*)^{(i)} \cdot x) = -1, B\}(|w^* \cdot x| + \gamma)/\gamma$$

$$\leq 2 \mathop{\mathbf{E}}_{(x,y)\sim D^{(i)}} (1 - \eta - \eta_i(x))\mathbb{1}\{\phi(w^* \cdot x, (w^*)^{(i)} \cdot x) = -1, |w^* \cdot x| < \gamma, B\}$$

$$\leq 2 \mathop{\mathbf{E}}_{(x,y)\sim D^{(i)}} (1 - 2\eta_i(x))\mathbb{1}\{\phi(w^* \cdot x, (w^*)^{(i)} \cdot x) = -1\}$$

We notice that

$$\mathrm{err}^{(i)}(w^*) = \mathop{\mathbf{E}}_{(x,y)\sim D^{(i)}} \eta_i(x) + \mathop{\mathbf{E}}_{(x,y)\sim D^{(i)}} (1 - 2\eta_i(x))\mathbb{1}\{\phi(w^* \cdot x, (w^*)^{(i)} \cdot x) = -1\}$$

$$= \mathrm{err}^{(i)}((w^*)^{(i)}) + \mathop{\mathbf{E}}_{(x,y)\sim D^{(i)}} (1 - 2\eta_i(x))\mathbb{1}\{\phi(w^* \cdot x, (w^*)^{(i)} \cdot x) = -1\}$$

This implies that

$$\mathop{\mathbf{E}}_{(x,y)\sim D^{(i)}} (1 - 2\eta_i(x))\mathbb{1}\{\phi(w^* \cdot x, (w^*)^{(i)} \cdot x) = -1\} = \mathrm{err}^{(i)}(w^*) - \mathrm{err}^{(i)}((w^*)^{(i)})$$

$$\leq \mathrm{err}^{(m+1)}(w^*) - \mathrm{err}^{(m+1)}((w^*)^{(i)}) + 2m\Delta \leq 2m\Delta.$$

Here, the first inequality follows the assumption on the drift rate, and the second inequality follows the optimality of $w^*$. This implies $\mathbf{E}_{(x,y)\sim D^{(i)}} U_i \mathbb{1}\{B\} \leq O(m\Delta)$. $\qquad\square$

## C.3 Proof of Lemma 3.5

We restate the lemma below.

**Lemma C.9** (Restatement of Lemma 3.5). *For $t, i > 0$ and $w^* \in \mathbb{S}^{d-1}$, define $\xi_i := \left(g(w^{(i)}; x, y) - \mathbf{E}_{(x,y)\sim D^{(t+i)}} g(w^{(i)}; x, y)\right) \cdot (w^{(i)} - w^*)$, $(x, y) \sim D^{(t+i)}$. For every $T > 0$, with probability at least $1 - \delta/2$, it holds $\sum_{i=1}^{T} \xi_i \leq \gamma^{-1}\sqrt{T}\log(1/\delta)$.*

*Proof of Lemma 3.5.* By Markov's inequality, we have

$$\mathbf{Pr}\left(\sum_{i=1}^{T} \xi_i \geq \alpha\right) = \mathbf{Pr}\left(\exp\left(\sum_{i=1}^{T} \xi_i\right) \geq \exp(\alpha)\right) \leq \mathbf{E}\exp\left(\sum_{i=1}^{T} \xi_i\right)\exp(-\alpha)$$

$$= \mathbf{E}\prod_{i=1}^{T}\exp(\xi_i \mid \mathcal{F}_i)\exp(-\alpha) \leq \exp\left(\sum_{t=1}^{T}\lambda_t^2/\gamma^2 - \alpha\right) \leq \delta.$$

Here, $\mathcal{F}_i, i \in [T]$ is the filtration. In the last inequality, we use the fact that $\xi_t$ is $O(1/\gamma)$-subgaussian and $\alpha = \sqrt{T}\log(1/\delta)/\gamma$. $\qquad\square$

## C.4 Omitted Details for Learning Drifting Halfspace under RCN

We will establish the following result.

**Theorem C.10** (Restatement of Theorem 3.6). *Consider the problem of learning drifting halfspaces with $\eta$-RCN. There is an algorithm $\mathcal{A}$ such that for any time step $T = \tilde{\Omega}(\Delta^{-2/3})$, $\mathcal{A}$ runs in $\mathrm{poly}(d, 1/\gamma, \Delta)$ time and outputs a hypothesis $\hat{h}^{(T)}$ such that with probability at least $9/10$, $\mathrm{err}^{(T)}(\hat{h}^{(T)}) \leq \mathrm{opt}_T + \tilde{O}(\Delta^{1/3}/\gamma)$.*

*Proof of Theorem 3.6.* To begin with, we show that within a certain window size $W$, the change of $\mathrm{opt}_t$ is small. Let $(w^*)^t$ and $(w^*)^{(t+1)}$ be the unit vectors that realize $D^{(t)}$ and $D^{(t+1)}$. Then we have

$$\mathrm{err}^{(t+1)}((w^*)^t) = (1 - \mathrm{opt}_{t+1})\mathop{\mathbf{Pr}}_{(x,y)\sim D^{(t+1)}} ((w^*)^t \cdot x)((w^*)^{t+1} \cdot x) < 0)$$

$$+ \mathrm{opt}_{t+1}\mathop{\mathbf{Pr}}_{(x,y)\sim D^{(t+1)}} ((w^*)^t \cdot x)((w^*)^{t+1} \cdot x) > 0)$$

$$= \mathrm{opt}_{t+1} + (1 - 2\mathrm{opt}_{t+1})\mathop{\mathbf{Pr}}_{(x,y)\sim D^{(t+1)}} ((w^*)^t \cdot x)((w^*)^{t+1} \cdot x) < 0)$$

$$\geq \mathrm{opt}_{t+1}.$$

This implies that $\text{opt}_{t+1} \le \text{opt}_t + \Delta$, otherwise $\text{err}^{(t+1)}((w^*)^t) \ge \text{err}^{(t)}((w^*)^t) + \Delta$, which gives a contradiction. By symmetry, we also have $\text{opt}_t \le \text{opt}_{t+1} + \Delta$. This implies $\left| \text{opt}_t - \text{opt}_{t+1} \right| \le \Delta$.

This implies that for any $t > 0$ and $T = \tilde{O}(\Delta^{-2/3})$, there is some $\eta$ such that for every $i \in T$, $\eta - \Delta T \le \text{opt}_{t+i} \le \eta$. This implies that by repeating Algorithm 1 with $\eta = \Delta T j/2$ for $j = 1, \ldots, 2(\Delta T)^{-1}$, we are able to get such a desired $\eta$. By Theorem 1.2, with the correct choice of $\eta$, we are able to get error $\text{err}^{(t)}(\hat{h}^{(t)}) \le \eta + \tilde{O}(\Delta^{1/3}/\gamma) \le \text{opt}_t + \tilde{O}(\Delta^{1/3}/\gamma)$. $\quad\square$

---

**Algorithm 4** DRIFTEDRCN (Learning Drifting Halfspaces with Random Classification Noise)

---

1: **Input:** $\eta \in (0, 1/2)$ : noise level, $\gamma \in (0, 1)$ : margin parameter, $\Delta > 0$ drift rate
2: **Output:** Predicted label $\hat{y}^{(t)} \in \{\pm 1\}$ for $x^{(t)}, t \ge 1$
3: Let $W = \tilde{\Theta}(\Delta^{-2/3}), i = 1, \hat{h}^0(x) \equiv 1$
4: **for** $t = 1, 2, \ldots$ **do**
5: $\quad$ Output hypothesis $h^{(i-1)}$ and receive $(x^{(t)}, y^{(t)})$.
6: $\quad$ $S^{(i)} \leftarrow S^{(i)} \cup \{(x^{(t)}, y^{(t)})\}$
7: $\quad$ **if** $t = iW$ **then**
8: $\quad\quad$ **for** $j \in [(\Delta W)^{-1}]$ **do**
9: $\quad\quad\quad$ $\eta_j = \Delta W j/2, h_j^{(i)} \leftarrow \text{DRIFTPERCEPTRON}(\eta_j, \gamma, S^{(i)}, \gamma/W)$
10: $\quad\quad$ $h^{(i)} \leftarrow \min \hat{\text{err}}(h_j^{(i)}), S^{(i)} \leftarrow \emptyset$
11: $\quad\quad$ $i \leftarrow i + 1$

---

## C.5 Proof of Theorem 3.7

We restate the theorem below.

**Theorem C.11** (Restatement of Theorem 3.7). *Consider the problem of learning drifting halfspaces. There is an algorithm $\mathcal{A}$ such that for any time step $T = \tilde{\Omega}((\gamma\Delta)^{-1/2})$, $\mathcal{A}$ runs in $\text{poly}(d, 1/\gamma, \Delta)$ time and outputs a hypothesis $\hat{h}^{(T)}$ such that with probability at least $9/10$, $\text{err}^{(T)}(\hat{h}^{(T)}) = \tilde{O}(\sqrt{\Delta}\gamma^{-3/2})$.*

---

**Algorithm 5** DRIFTEDHALFSPACE (Learning Drifting Realizable Halfspaces)

---

1: **Input:** $\gamma \in (0, 1)$ : margin parameter, $\Delta > 0$ drift rate
2: **Output:** Predicted label $\hat{y}^{(t)} \in \{\pm 1\}$ for $x^{(t)}, t \ge 1$
3: Let $W = \tilde{\Theta}((\gamma\Delta)^{-1/2}), i = 1, \hat{h}^0(x) \equiv 1$
4: **for** $t = 1, 2, \ldots$ **do**
5: $\quad$ Output hypothesis $h^{(i-1)}$ and receive $(x^{(t)}, y^{(t)})$
6: $\quad$ $S^{(i)} \leftarrow S^{(i)} \cup \{(x^{(t)}, y^{(t)})\}$
7: $\quad$ **if** $t = iW$ **then**
8: $\quad\quad$ $h^{(i)} \leftarrow \text{REALIZABLEPERCEPTRON}(\gamma, S^{(i)}, \gamma), S^{(i)} \leftarrow \emptyset$
9: $\quad\quad$ $i \leftarrow i + 1$

---

**Algorithm 6** REALIZABLEPERCEPTRON (Subroutine for Learning a Single Halfspace over a Dataset)

---

1: **Input:** $\gamma \in (0, 1)$ : margin parameter, $S = \{(x^{(i)}, y^{(i)})\}_{i=1}^m$ a dataset of $2m$ labeled examples, $\mu > 0$ step size,
2: **Output:** $\hat{h} = \text{sign}(\hat{w} \cdot x) : \mathbb{R}^d \to \{\pm 1\}$
3: $w^{(0)} = e_1$
4: Split $S = S_1 \cup S_2$, where $S_1 = \{(x^{(i)}, y^{(i)})\}_{i=1}^{m/2}, S_2 = S \setminus S_1$
5: **for** $i = 1, 2, \ldots, m$ **do**
6: $\quad$ $g(w^{(i)}, x, y) \leftarrow (\text{sign}(w^{(i)} \cdot x) - y)x$
7: $\quad$ $w^{(i+1)} = \text{proj}_{\mathbb{B}(1)}(w^{(i)} - \mu g_t(w^{(i)}, x, y)), \hat{h}_{i+1}(x) = \text{sign}(w^{(i+1)} \cdot x)$
8: Return $\hat{h} = \text{argmin}_{\hat{h}_i} \hat{\text{err}}(\hat{h}_i)$, where $\hat{\text{err}}(\hat{h}_i) = \frac{1}{|S_2|} \sum_{(x,y) \in S_2} \mathbb{1}\{\hat{h}_i(x) \ne y\}$.

---

*Proof of Theorem 3.7.* We will show that for $t = iW$, $\mathrm{err}^{(t)}(h^{(i)}) \leq \tilde{O}(\Delta^{1/2}/\gamma^{3/2})$. Suppose this is correct, then for $t = iW + j, j \leq W$, we have

$$\mathrm{err}^{(t)}(h^{(i)}) = \Pr_{(x,y)\sim D^{(t)}}(h^{(i)}(x) \neq y) \leq \Pr_{(x,y)\sim D^{(iW)}}(h^{(i)}(x) \neq y) + j\Delta$$

$$\leq \tilde{O}(\Delta^{1/2}/\gamma^{3/2}) + \Delta^{1/2}/\gamma = \tilde{O}(\Delta^{1/2}/\gamma^{3/2}).$$

To do this, we will analyze Algorithm 6 with $t = iW$. Let $w^{(i)}$ be the vector in the $i$th round of update in Algorithm 6 and let $w^*$ be the unit vector that realizes $D^{(t+T)}$, where $T = W/2$. By the contraction of the projection operator, we have

$$\left\| w^{(i+1)} - w^* \right\|^2 \leq \left\| w^{(i)} - w^* - \mu g(w^{(i)}; x, y) \right\|^2$$

$$\leq \left\| w^{(i)} - w^* \right\|^2 + \mu^2 \left\| g(w^{(i)}; x, y) \right\|^2 - 2\mu g(w^{(i)}; x, y) \cdot (w^{(i)} - w^*)$$

$$= \left\| w^{(i)} - w^* \right\|^2 + \mu^2 \mathbb{1}\{h_{w^{(i)}}(x) \neq y\} + 2\mu \mathbb{1}\{h_{w^{(i)}}(x) \neq y\}yx \cdot (w^{(i)} - w^*)$$

Rearrange the inequality we obtain that

$$\mathbb{1}\{h_{w^{(i)}}(x) \neq y\} \left( 2\mu yx \cdot (w^* - w^{(i)}) - \mu^2 \right) \leq \left\| w^{(i)} - w^* \right\|^2 - \left\| w^{(i+1)} - w^* \right\|^2,$$

which implies

$$\mathbb{1}\{h_{w^{(i)}}(x) \neq y\} \left( 2\mu yx \cdot w^* - \mu^2 \right) \leq \left\| w^{(i)} - w^* \right\|^2 - \left\| w^{(i+1)} - w^* \right\|^2$$

Denote by $I_i := \mathbb{1}\{h_{w^{(i)}}(x) \neq y\} \left( 2\mu yx \cdot w^* - \mu^2 \right)$ for $i \in [T]$. We notice that for every $w^{(i)} \in \mathbb{S}^{d-1}$,

$$\mathbb{E}_{(x,y)\sim D^{(t+i)}} I_i = \mathbb{E}_{(x,y)\sim D^{(t+i)}} \mathbb{1}\{h_{w^{(i)}}(x) \neq y\}\mathbb{1}\{yx \cdot w^* \geq \gamma\} \left( 2\mu yx \cdot w^* - \mu^2 \right)$$

$$+ \mathbb{E}_{(x,y)\sim D^{(t+i)}} \mathbb{1}\{h_{w^{(i)}}(x) \neq y\}\mathbb{1}\{yx \cdot w^* < \gamma\} \left( 2\mu yx \cdot w^* - \mu^2 \right)$$

$$\geq \mathbb{E}_{(x,y)\sim D^{(t+i)}} \mathbb{1}\{h_{w^{(i)}}(x) \neq y\}\mathbb{1}\{yx \cdot w^* \geq \gamma\}\mu(2\gamma - \mu)$$

$$+ \mathbb{E}_{(x,y)\sim D^{(t+i)}} \mathbb{1}\{h_{w^{(i)}}(x) \neq y\}\mathbb{1}\{yx \cdot w^* < \gamma\} \left( 2\mu yx \cdot w^* - \mu^2 \right)$$

$$= \mathbb{E}_{(x,y)\sim D^{(t+i)}} \mathbb{1}\{h_{w^{(i)}}(x) \neq y\}\mu(2\gamma - \mu)$$

$$- \mathbb{E}_{(x,y)\sim D^{(t+i)}} \mathbb{1}\{h_{w^{(i)}}(x) \neq y\}\mathbb{1}\{yx \cdot w^* < \gamma\}\mu(2\gamma - \mu)$$

$$+ \mathbb{E}_{(x,y)\sim D^{(t+i)}} \mathbb{1}\{h_{w^{(i)}}(x) \neq y\}\mathbb{1}\{yx \cdot w^* < \gamma\} \left( 2\mu yx \cdot w^* - \mu^2 \right)$$

We upper bound the three terms separately. For the first term, we have

$$\mathbb{E}_{(x,y)\sim D^{(t+i)}} \mathbb{1}\{h_{w^{(i)}}(x) \neq y\}\mu(2\gamma - \mu) \geq \mu\gamma\mathrm{err}^{(t+i)}(w^{(i)}),$$

when $\mu \leq \gamma$. For the second term, according to the drift rate assumption, we have

$$\mathbb{E}_{(x,y)\sim D^{(t+i)}} \mathbb{1}\{h_{w^{(i)}}(x) \neq y\}\mathbb{1}\{yx \cdot w^* < \gamma\}\mu(2\gamma - \mu) \leq 2\Delta W\mu\gamma$$

For the third term, we have

$$\mathbb{E}_{(x,y)\sim D^{(t+i)}} \mathbb{1}\{h_{w^{(i)}}(x) \neq y\}\mathbb{1}\{yx \cdot w^* < \gamma\} \left( 2\mu yx \cdot w^* - \mu^2 \right) \geq -\Delta W\mu.$$

This implies $\mathbb{E}_{(x,y)\sim D^{(t+i)}} I_i \geq \mu\gamma\mathrm{err}^{(t+i)}(w^{(i)}) - 2\Delta W\mu$.

By taking a summation for $i \in [T]$, we have

$$\frac{1}{T}\sum_{i=1}^{T}\mathrm{err}^{(t+i)}(w^{(i)}) \le \frac{4}{\mu\gamma T} + \frac{1}{\mu\gamma T}\sum_{i=1}^{T}(\mathop{\mathbf{E}}_{(x,y)\sim D^{(t+i)}} I_i - I_i) + 2\frac{\Delta T}{\gamma} \le 2\Delta^{1/2}\gamma^{-2/3} + \frac{1}{\mu\gamma T}\sum_{i=1}^{T}(\mathop{\mathbf{E}}_{(x,y)\sim D^{(t+i)}} I_i - I_i).$$

$$(13)$$

Here, the second inequality follows the choice of $\mu = \gamma$ and $T = (\gamma\Delta)^{-1/2}$. It remains to analyze the term $\frac{1}{\mu\gamma T}\sum_{i=1}^{T}(\mathbf{E}_{(x,y)\sim D^{(t+i)}} I_i - I_i)$. To do this, we upper bound the variance of $\mathbf{E}_{(x,y)\sim D^{(t+i)}} I_i - I_i$, which is at most We have

$$\begin{aligned}
\mathop{\mathbf{E}}_{(x,y)\sim D^{(t+i)}} I_i^2 &= \mathop{\mathbf{E}}_{(x,y)\sim D^{(t+i)}} \mathbb{1}\{h_{w^{(i)}}(x) \ne y\}\mathbb{1}\{yx\cdot w^* \ge \gamma\}\left(2\mu yx\cdot w^* - \mu^2\right)^2 \\
&\quad + \mathop{\mathbf{E}}_{(x,y)\sim D^{(t+i)}} \mathbb{1}\{h_{w^{(i)}}(x) \ne y\}\mathbb{1}\{yx\cdot w^* < \gamma\}\left(2\mu yx\cdot w^* - \mu^2\right)^2 \\
&\le \mu^2\mathrm{err}^{(t+i)}(w^{(i)}).
\end{aligned}$$

Thus, by Bernstein's inequality, we have with probability at least $1 - \delta$,

$$\sum_{i=1}^{T}\mathop{\mathbf{E}}_{(x,y)\sim D^{(t+i)}} I_i - I_i \le O\left(\log(1/\delta) + \sqrt{\sum_{i=1}^{T}\mu^2\mathrm{err}^{(t+i)}(w^{(i)})}\right). \qquad (14)$$

Combining (13) and (14), we have with probability at least $1 - \delta$, $\frac{1}{T}\sum_{i=1}^{T}\mathrm{err}^{(t+i)}(w^{(i)}) \le \tilde{O}(\Delta^{1/2}\gamma^{-2/3})$. It remains to show that we are able to select a good hypothesis by looking at the empirical error.

For each $j \in [T]$, we consider the halfspace $\hat{h}_j = h_{w^{(j)}}$. By the assumption on the drift rate, we have $\left|\mathrm{err}^{((i+1)W)}(\hat{h}_j) - \mathrm{err}^{(t+\ell)}(\hat{h}_j)\right| \le \Delta W = \Delta^{1/2}\gamma^{-1/2}$ for every $\ell \in [W]$, which implies that

$$\left|\mathrm{err}^{((i+1)W)}(\hat{h}_j) - \frac{2}{W}\sum_{\ell=1}^{W/2}\mathrm{err}^{(t+\ell+W/2)}(\hat{h}_j)\right| \le \Delta W.$$

By Hoeffding's inequality, we have

$$\mathbf{Pr}\left(\hat{\mathrm{err}}(\hat{h}_j) - \frac{2}{W}\sum_{\ell=1}^{W/2}\mathrm{err}^{(t+\ell+W/2)}(\hat{h}_j) > \mathrm{err}^{((i+1)W)}(\hat{h}_j)\right) \le \delta.$$

Thus, with probability at least $1 - \delta$, we have $\left|\hat{\mathrm{err}}(\hat{h}_j) - \mathrm{err}^{((i+1)W)}(\hat{h}_j)\right| \le \mathrm{err}^{((i+1)W)}(\hat{h}_j)$. Since there is some $j^* \in [T]$ such that $\mathrm{err}^{(t+j^*)}(w^{(j^*)}) \le \tilde{O}(\Delta^{1/2}\gamma^{-3/2})$, we know that $\mathrm{err}^{((i+1)W)}(w^{(j^*)}) \le \tilde{O}(\Delta^{1/2}\gamma^{-3/2})$. This implies the selected hypothesis that minimizes $\hat{\mathrm{err}}(h_j)$ must have $\mathrm{err}^{((i+1)W)}(\hat{h}) \le \tilde{O}(\Delta^{1/2}\gamma^{-3/2})$. $\qquad \square$

# D  Omitted Proofs from Section 4

## D.1  Proof of Lemma D.1

**Lemma D.1.** *The instance constructed in Definition 4.3 is a valid instance of the trajectory testing problem defined in Definition 4.1 with $\eta = 1/3$.*

*Proof of Lemma D.1.* To show the construction in Definition 4.3 is a valid instance, we will need to show that for each $v$ and $i \in [T]$, the defined halfspaces $h_i^{(v)}$ has margin $\gamma$ and for every $i \in [T]$, $d_{TV}(D_i^{(v)}, D_{i+1}^{(v)}) \le \Delta$.

We start with analyzing the margin. Let $h_i^{(v)} = \mathrm{sign}(v \cdot x + t_i)$, since $v \in \{\pm 1\}^d z$, $x \in \{\pm 1\}$ and $d$ is odd, we have $|v \cdot x| \ge 1$. Since $(D_i^{(v)})_x$ is the uniform distribution over $\{\pm 1\}^d$, we know that for any choice of $t_i$, there is an integer $t_i'$

such that for every $x \in \{\pm 1\}^d$, $|v \cdot x + t_i'| \geq 1$, and

$$\Pr_{D_i^{(v)}} \left(\text{sign}(v \cdot x + t_i) = -1\right) = \Pr_{D_i^{(v)}} \left(\text{sign}(v \cdot x + t_i') = -1\right)$$

This implies that by setting $t_i = t_i'$, halfspace $\text{sign}(v \cdot x + t_i')$ has a margin $\Omega(1/d) = \Omega(\gamma)$.

It remains to show that for every $i \geq T - m$, we have $d_{TV}(D_i^{(v)}, D_{i+1}^{(v)}) \leq \Delta$. Let $A \subseteq \{\pm 1\}^d \times \{\pm 1\}$ be any event. We have

$$\Pr_{D_i^{(v)}}(A) = \Pr_{D_i^{(v)}}(A, h_i^{(v)}(x) \neq h_i^{(v)}(x)) + \Pr_{D_i^{(v)}}(A, h_i^{(v)}(x) = h_i^{(v)}(x))$$
$$= \Pr_{D_i^{(v)}}(A, h_i^{(v)}(x) \neq h_i^{(v)}(x)) + \Pr_{D_i^{(v)}}(A, h_i^{(v)}(x) = h_i^{(v)}(x) = y(x)) + \Pr_{D_i^{(v)}}(A, h_i^{(v)}(x) = h_i^{(v)}(x) \neq y(x)).$$

This implies that

$$\left| \Pr_{D_i^{(v)}}(A) - \Pr_{D_{i+1}^{(v)}}(A) \right| \leq \left| \Pr_{D_i^{(v)}} - \Pr_{D_{i+1}^{(v)}}(A, h_i^{(v)}(x) \neq h_i^{(v)}(x)) \right| + \left| \Pr_{D_i^{(v)}} - \Pr_{D_{i+1}^{(v)}}(A, h_i^{(v)}(x) = h_i^{(v)}(x) = y(x)) \right|$$

$$+ \left| \Pr_{D_i^{(v)}} - \Pr_{D_{i+1}^{(v)}}(A, h_i^{(v)}(x) = h_i^{(v)}(x) \neq y(x)) \right|$$

$$\leq \left| \Pr_{D_i^{(v)}} - \Pr_{D_{i+1}^{(v)}}(h_i^{(v)}(x) \neq h_i^{(v)}(x)) \right| + \left| \Pr_{D_i^{(v)}} - \Pr_{D_{i+1}^{(v)}}(h_i^{(v)}(x) = h_i^{(v)}(x) = y(x)) \right|$$

$$+ \left| \Pr_{D_i^{(v)}} - \Pr_{D_{i+1}^{(v)}}(A, h_i^{(v)}(x) = h_i^{(v)}(x) \neq y(x)) \right|$$

$$\leq \Delta + 2((\eta - \Delta(j+1))/(1 - 2\Delta(j+1)) - (\eta - \Delta j)/(1 - 2\Delta j))$$
$$\leq 4\Delta.$$

To conclude the proof of Lemma D.1, it remains to show that we can actually make $\Pr_{D_i^{(v)}}(h_i^{(v)}(x) = -1) = \Delta j$ by showing that changing the threshold $t$ will not make $\Pr_{D_i^{(v)}}(h_i^{(v)}(x) = -1)$ dramatically change. Since $v \in \{\pm 1\}^d$ and $x \in \{\pm 1\}^d$, we know that $v \cot x \in [-d, d]$ is an integer. Since $(D_i^{(v)})_x$ is a uniform distribution over $\{\pm 1\}$, we know that for every integer $c$,

$$\Pr_{(D_i^{(v)})_x} (v \cdot x = c) \leq \binom{d}{d/2}/2^d \leq 2^{-O(d)}.$$

This implies that as long as $\gamma < O(\log^{-1}(1/\Delta))$, we can make $\Pr_{(D_i^{(v)})_x}(v \cdot x = c) \leq \Delta/100$. This implies that we are able to choose $h_i^{(v)}$ such that $\Pr_{D_i^{(v)}}(h_i^{(v)}(x) = -1) = \Delta j$. $\qquad \square$

## D.2 Proof of Lemma 4.6

**Lemma D.2** (Restatement of Lemma 4.6). *Under the construction of Definition 4.3, we have*

$$\langle (D^{\bar{(u)}\leq \ell, k}), (D^{\bar{(v)}\leq \ell, k}) \rangle_{D^{(0)}} - 1 = \sum_{A \subseteq [m], A \neq \emptyset, |A| \leq k} \prod_{j \in [A]} \langle (D_{T-m+j}^{\bar{(u)}})^{\leq \ell}(z_i)), (D_{T-m+j}^{\bar{(v)}})^{\leq \ell}(z_i) \rangle_{D_i^{(0)}} - 1$$

*Proof of Lemma 4.6.* Since $D_i^{(u)}$ According to the construction of $D^{(0)}$ and $\mathcal{D}$, for every $u \in \{\pm 1\}^d$, we have $D^{\bar{(u)}} =$

$\prod_{i=1}^{T} D_i^{(u)}(z_i) = \prod_{i=T-m+1}^{T} D_i^{(u)}(z_i)$. This implies that

$$
\begin{aligned}
D^{\overline{(u)}\leq\ell,k}(z) &= \left(\prod_{i=T-m+1}^{T} D_i^{\overline{(u)}}(z_i)\right)^{\leq\ell,k} = \left(\prod_{j=1}^{m} D_{T-m+j}^{(u^{\overline{}})}(z_i)\right)^{\leq\ell,k} \\
&= \left(\prod_{j=1}^{m} 1 + (D_{T-m+j}^{(u^{\overline{}})})^{\leq\ell}(z_i) - 1 + (D_{T-m+j}^{(u^{\overline{}})})^{>\ell}(z_i)\right)^{\leq\ell,k} \\
&= \sum_{A\subseteq[m],B\subseteq[m]\backslash A} \left(\prod_{j\in[A]} ((D_{T-m+j}^{(u^{\overline{}})})^{\leq\ell}(z_i) - 1) \prod_{j\in[m]\backslash(A\cup B)} (D_{T-m+j}^{(u^{\overline{}})})^{>\ell}(z_i)\right)^{\leq\ell,k} \\
&= \sum_{A\subseteq[m]} \left(\prod_{j\in[A]} ((D_{T-m+j}^{(u^{\overline{}})})^{\leq\ell}(z_i) - 1)\right)^{\leq\ell,k} \\
&= \sum_{A\subseteq[m],|A|\leq k} \left(\prod_{j\in[A]} ((D_{T-m+j}^{(u^{\overline{}})})^{\leq\ell}(z_i) - 1)\right)^{\leq\ell,k}
\end{aligned}
$$

Here, the second last equation follows $\left(\prod_{j\in[A]}((D_{T-m+j}^{(u^{\overline{}})})^{\leq\ell}(z_i) - 1)\prod_{j\in[m]\backslash(A\cup B)}(D_{T-m+j}^{(u^{\overline{}})})^{>\ell}(z_i)\right)^{\leq\ell,k} = 0$, when $[m]\backslash(A\cup B) \neq \emptyset$. In the last equation, we use the fact that $\left(\prod_{j\in[A]}((D_{T-m+j}^{(u^{\overline{}})})^{\leq\ell}(z_i) - 1)\right)^{\leq\ell,k} = 0$, when $|A| > k$. Furthermore, let $A, B \subseteq [m]$ and $A \neq B$, by independence, we have

$$
\langle\left(\prod_{j\in[A]} ((D_{T-m+j}^{(u^{\overline{}})})^{\leq\ell,k}(z_i) - 1)\right)^{\leq\ell}, \left(\prod_{j\in[B]} ((D_{T-m+j}^{(v^{\overline{}})})^{\leq\ell}(z_i) - 1)\right)^{\leq\ell,k}\rangle_{D^{(0)}} = 0
$$

This implies

$$
\begin{aligned}
&\langle(D^{\overline{(u)}\leq\ell,k}), (D^{\overline{(v)}\leq\ell,k})\rangle_{D^{(0)}} - 1 \\
&= \langle \sum_{A\subseteq[m],|A|\leq k} \left(\prod_{j\in[A]} ((D_{T-m+j}^{(u^{\overline{}})})^{\leq\ell}(z_i) - 1)\right)^{\leq\ell,k}, \sum_{A\subseteq[m]} \left(\prod_{j\in[A]} ((D_{T-m+j}^{(v^{\overline{}})})^{\leq\ell}(z_i) - 1)\right)^{\leq\ell,k}\rangle_{D^{(0)}} - 1 \\
&= \sum_{A\subseteq[m],|A|\leq k} \langle \left(\prod_{j\in[A]} ((D_{T-m+j}^{(u^{\overline{}})})^{\leq\ell}(z_i) - 1)\right)^{\leq\ell,k}, \left(\prod_{j\in[A]} ((D_{T-m+j}^{(v^{\overline{}})})^{\leq\ell}(z_i) - 1)\right)^{\leq\ell,k}\rangle_{D^{(0)}} - 1 \\
&= \sum_{A\subseteq[m],|A|\leq k} \langle \prod_{j\in[A]} ((D_{T-m+j}^{(u^{\overline{}})})^{\leq\ell}(z_i) - 1), \prod_{j\in[A]} ((D_{T-m+j}^{(v^{\overline{}})})^{\leq\ell}(z_i) - 1)\rangle_{D^{(0)}} - 1 \\
&= \sum_{A\subseteq[m],A\neq\emptyset,|A|\leq k} \prod_{j\in[A]} \langle ((D_{T-m+j}^{(u^{\overline{}})})^{\leq\ell}(z_i) - 1), ((D_{T-m+j}^{(v^{\overline{}})})^{\leq\ell}(z_i) - 1)\rangle_{D_i^{(0)}} \\
&= \sum_{A\subseteq[m],A\neq\emptyset,|A|\leq k} \prod_{j\in[A]} \langle (D_{T-m+j}^{(u^{\overline{}})})^{\leq\ell}(z_i)), (D_{T-m+j}^{(v^{\overline{}})})^{\leq\ell}(z_i)\rangle_{D_i^{(0)}} - 1.
\end{aligned}
$$

Here, the third equation follows the fact that $\prod_{j\in[A]}((D_{T-m+j}^{(u^{\overline{}})})^{\leq\ell}(z_i) - 1) \in \mathcal{C}_{\ell,k}$ and the last equation follows the independence of each $z_i$. $\qquad\square$

With Lemma 4.6, it remains to bound the pairwise correlation.

### D.3 Proof of Lemma 4.7

We restate the lemma below.

**Lemma D.3** (Restatement of Lemma 4.7). *Let $c \in (0, 1/2)$ be a constant. For $i = T - m + j, j \in [m]$ and $v, u \in \{\pm 1\}^d$ such that $|v \cdot u| \leq d^{1-c}$, we have*

$$\langle D^{(u)^-}_{T-m+j}(z_i), D^{(v)^-}_{T-m+j}(z_i) \rangle_{D^{(0)}_i} - 1 \leq \tilde{O}(d^{-(1/2-c)}(\Delta j)^2).$$

*Proof of Lemma 4.7.* Recall that in Definition 4.3, for every $i = T - m + j, j \in [m]$ and for every $u \in \{\pm 1\}^d$, $D^{(u)}$ is realized by halfspaces $h^{(u)}_i = \text{sign}(v \cdot x + t_i)$ with RCN $\eta_i = (\eta - \Delta j)/(1 - 2\Delta j)$. In particular, $\mathbf{Pr}_{D^{(v)}_i}(h^{(v)}_i(x) = -1) = \Delta j$. This implies that

$$\mathbf{Pr}_{D^{(v)}_i}(y(x) = -1) = (1 - \eta_i) \mathbf{Pr}_{D^{(v)}_i}(h^{(v)}_i(x) = -1) + \eta_i \mathbf{Pr}_{D^{(v)}_i}(h^{(v)}_i(x) = +1)$$

$$= \eta_i + (1 - 2\eta_i) \mathbf{Pr}_{D^{(v)}_i}(h^{(v)}_i(x) = -1)$$

$$= \frac{\eta - \Delta j}{1 - 2\Delta j} + (1 - 2\frac{\eta - \Delta j}{1 - 2\Delta j})\Delta j = \eta.$$

Since $D^{(0)}_i$ is a product distribution over $\{\pm 1\}^d \times \{\pm 1\}$ and $(D^{(0)}_i)_x$ is the uniform distribution over $\{\pm 1\}^d$, we have

$$\langle D^{(u)^-}_i(z_i), D^{(v)^-}_i(z_i) \rangle_{D^{(0)}_i} - 1$$

$$= \sum_{y \in \{\pm 1\}} \sum_{x \in \{\pm 1\}^d} D^{(u)}_i(x, y) D^{(v)}_i(x, y)/D^{(0)}_i(x, y) - 1$$

$$= \eta \left( \sum_{x \in \{\pm 1\}^d} \frac{D^{(u)}_i(x)|_{y=-1} D^{(v)}_i(x)|_{y=-1}}{D^{(0)}_i(x)|_{y=-1}} - 1 \right) + (1 - \eta) \left( \sum_{x \in \{\pm 1\}^d} \frac{D^{(u)}_i(x)|_{y=1} D^{(v)}_i(x)|_{y=1}}{D^{(0)}_i(x)|_{y=1}} - 1 \right)$$

$$= \eta \left( \sum_{x \in \{\pm 1\}^d} \frac{D^{(u)}_i(x)|_{y=-1} D^{(v)}_i(x)|_{y=-1}}{D^{(0)}_i(x)} - 1 \right) + (1 - \eta) \left( \sum_{x \in \{\pm 1\}^d} \frac{D^{(u)}_i(x)|_{y=1} D^{(v)}_i(x)|_{y=1}}{D^{(0)}_i(x)} - 1 \right)$$

Notice that

$$D^{(v)}_i(x)|_{y=-1} = \left( \mathbb{1}\{h^{(v)}_i(x) = 1\}\eta_i D^{(0)}_i(x) + \mathbb{1}\{h^{(v)}_i(x) = -1\}(1 - \eta_i) D^{(0)}_i(x) \right)/\eta$$

$$= \left( \eta_i + \mathbb{1}\{h^{(v)}_i(x) = -1\}(1 - 2\eta_i) \right) D^{(0)}_i(x)/\eta$$

$$D^{(v)}_i(x)|_{y=1} = \left( \mathbb{1}\{h^{(v)}_i(x) = -1\}\eta_i D^{(0)}_i(x) + \mathbb{1}\{h^{(v)}_i(x) = 1\}(1 - \eta_i) D^{(0)}_i(x) \right)/(1 - \eta)$$

$$= \left( 1 - \eta_i - \mathbb{1}\{h^{(v)}_i(x) = -1\}(1 - 2\eta_i) \right) D^{(0)}_i(x)/(1 - \eta).$$

This implies that

$$\langle D^{(u)^-}_i(x)|_{y=-1}, D^{(v)^-}_i(x)|_{y=-1} \rangle_{D^{(0)}_i} - 1$$

$$= \langle \eta_i + \mathbb{1}\{h^{(v)}_i(x) = -1\}(1 - 2\eta_i), \eta_i + \mathbb{1}\{h^{(v)}_i(x) = -1\}(1 - 2\eta_i) \rangle_{D^{(0)}_i}/\eta^2 - 1$$

$$= (\eta_i^2 + 2\eta_i(1 - 2\eta_i) \mathbf{Pr}_{D^{(v)}_i}(h^{(v)}_i(x) = -1) + (1 - 2\eta_i)^2 \mathbf{E}_{D^{(0)}_i}(\mathbb{1}\{h^{(v)}_i(x) = -1\}\mathbb{1}\{h^{(u)}_i(x) = -1\}))\}/\eta^2 - 1$$

$$= \left( \frac{1 - 2\eta_i}{\eta} \right)^2 \mathbf{E}_{D^{(0)}_i} \left( (\mathbb{1}\{h^{(v)}_i(x) = -1\} - \Delta j)(\mathbb{1}\{h^{(u)}_i(x) = -1\} - \Delta j) \right)$$

and

$$\langle \bar{D}_i^{(u)}(x)\mid_{y=1}, \bar{D}_i^{(v)}(x)\mid_{y=1}\rangle_{D_i^{(0)}} - 1$$

$$= \langle 1 - \eta_i - \mathbb{1}\{h_i^{(v)}(x) = -1\}(1-2\eta_i), 1 - \eta_i - \mathbb{1}\{h_i^{(v)}(x) = -1\}(1-2\eta_i)\rangle_{D_i^{(0)}}/(1-\eta)^2 - 1$$

$$= ((1-\eta_i)^2 - 2(1-\eta_i)(1-2\eta_i)\Pr_{D_i^{(v)}}(h_i^{(v)}(x) = -1) + (1-2\eta_i)^2 \mathop{\mathbf{E}}_{D_i^{(0)}}(\mathbb{1}\{h_i^{(v)}(x) = -1\}\mathbb{1}\{h_i^{(u)}(x) = -1\}))\}/(1-\eta)^2 - 1$$

$$= \left(\frac{1-2\eta_i}{1-\eta}\right)^2 \mathop{\mathbf{E}}_{D_i^{(0)}}\left((\mathbb{1}\{h_i^{(v)}(x) = -1\} - \Delta j)(\mathbb{1}\{h_i^{(u)}(x) = -1\} - \Delta j)\right).$$

This gives

$$\langle \bar{D}_i^{(u)}(z_i), \bar{D}_i^{(v)}(z_i)\rangle_{D_i^{(0)}} - 1 = (1-2\eta_i)^2 \mathop{\mathbf{E}}_{D_i^{(0)}}\left((\mathbb{1}\{h_i^{(v)}(x) = -1\} - \Delta j)(\mathbb{1}\{h_i^{(u)}(x) = -1\} - \Delta j)\right)\left(\frac{1}{\eta} + \frac{1}{1-\eta}\right)$$

$$\leq O\left(\mathop{\mathbf{E}}_{D_i^{(0)}}\left((\mathbb{1}\{h_i^{(v)}(x) = -1\}(\mathbb{1}\{h_i^{(u)}(x) = -1\} - (\Delta j)^2))\right)\right).$$

*Fact* D.4 (Lemma 3.5 in (Diakonikolas et al., 2023)). For $v, u \in \{\pm 1\}^d$ such that $|v \cdot u| \leq d^{1-c}$ and halfspace $f_v(x), f_u(x)$ satisfies $\mathbf{Pr}_{x\sim\{\pm1\}^d}(f_v(x) = 1) = \mathbf{Pr}_{x\sim\{\pm1\}^d}(f_v(x) = 1) = \epsilon$, $\epsilon \in (0,1)$ small enough, there is some absolute constant $C > 0$ such that $\mathbf{E}_{x\sim\{\pm1\}^d}\mathbb{1}\{f_v(x) = 1\}\mathbb{1}\{f_u(x) = 1\} \leq C\log^2(d/\epsilon)\epsilon^2\,|v \cdot u|\,/d + \epsilon^2$.

Applying Fact D.4 to $-h_i^{(v)}, -h_i^{(u)}$ with $\epsilon = \Delta j$, we have

$$\mathop{\mathbf{E}}_{D_i^{(0)}}\left((\mathbb{1}\{h_i^{(v)}(x) = -1\}(\mathbb{1}\{h_i^{(u)}(x) = -1\} - (\Delta j)^2))\right) \leq \tilde{O}(d^{-(1/2-c)}(\Delta j)^2).$$

$\square$

## D.4  Proof of Theorem 4.4

We restate the theorem below.

**Theorem D.5** (Restatement of Theorem 4.4). *Let $c \in (0, 1/2)$ be a constant. For $\eta = 1/3$ and $\Delta > 2^{-1/\gamma^c}$, there is no polynomial $p(z)$ with degree less than $O(\gamma^{-c/4})$ that is a 1-distinguisher for the trajectory testing problem defined in Definition 4.1, with instance constructed by Definition 4.3.*

*Proof of Theorem 4.4.* By Fact 4.5, to show there is no polynomial with degree $k = O(d^{c/4})$ that is a $O(1)$-distinguisher for $D^{(0)}$ and $D$, it suffices to upper bound $\left\|\mathbf{E}_{D\sim\mathcal{D}}(\bar{D}^{\leq\infty,k}) - 1\right\|_{D^{(0)}}$ for $D^{(0)}$ and $\mathcal{D}$ constructed in Definition 4.3. We have

$$\left\|\mathop{\mathbf{E}}_{D\sim\mathcal{D}}(\bar{D}^{\leq\infty,k}) - 1\right\|^2_{D^{(0)}} = \langle \mathop{\mathbf{E}}_{D^{(u)}\sim\mathcal{D}}(\bar{D^{(u)}}^{\leq\infty,k}) - 1, \mathop{\mathbf{E}}_{D^{(v)}\sim\mathcal{D}}(\bar{D^{(v)}}^{\leq\infty,k}) - 1\rangle_{D^{(0)}}$$

$$= \langle \mathop{\mathbf{E}}_{D^{(u)}\sim\mathcal{D}}(\bar{D^{(u)}}^{\leq\infty,k}), \mathop{\mathbf{E}}_{D^{(v)}\sim\mathcal{D}}(\bar{D^{(v)}}^{\leq\infty,k})\rangle_{D^{(0)}} - 1$$

$$= \mathop{\mathbf{E}}_{D^{(u)},D^{(v)}\sim\mathcal{D}}\langle (\bar{D^{(u)}}^{\leq\infty,k}), (\bar{D^{(v)}}^{\leq\infty,k})\rangle_{D^{(0)}} - 1,$$

where in the second equality, we use the fact that $\langle \bar{D^{(u)}}^{\leq\infty,k}, 1\rangle_{D^{(0)}} = 1$. By Lemma 4.6, we have

$$\mathop{\mathbf{E}}_{D^{(u)},D^{(v)}\sim\mathcal{D}}\langle (\bar{D^{(u)}}^{\leq\infty,k}), (\bar{D^{(v)}}^{\leq\infty,k})\rangle_{D^{(0)}} - 1$$

$$= \sum_{A\subseteq[m],A\neq\emptyset,|A|\leq k} \mathop{\mathbf{E}}_{D^{(u)},D^{(v)}\sim\mathcal{D}}\prod_{j\in[A]}\left(\langle D_{T-m+j}^{(u)\bar{}}(z_i)), D_{T-m+j}^{(v)\bar{}}(z_i)\rangle_{D_i^{(0)}} - 1\right).$$

Thus, to conclude the proof, it remain to upper bound

$$\mathop{\mathbf{E}}_{D^{(u)},D^{(v)}\sim\mathcal{D}}\prod_{j\in[A]}\left(\langle D_{T-m+j}^{(u)\bar{}}(z_i)), D_{T-m+j}^{(v)\bar{}}(z_i)\rangle_{D_i^{(0)}} - 1\right).$$

We have

$$\underset{D^{(u)}, D^{(v)} \sim \mathcal{D}}{\mathbf{E}} \prod_{j \in [A]} \left( \langle D_{T-m+j}^{(u)}(z_i)), D_{T-m+j}^{(v)}(z_i) \rangle_{D_i^{(0)}} - 1 \right)$$

$$= \frac{1}{|S|} \prod_{j \in [A]} (\Delta j) + (1 - \frac{1}{|S|}) \prod_{j \in [A]} (d^{-(1/2-c)}(\Delta j)^2)$$

$$\leq 2^{-\Omega(d^c)}(\Delta m)^{|A|} + (d^{-(1/2-c)}(\Delta m)^2)^{|A|}$$

$$\leq 2^{-\Omega(d^c)}(d^{1/6}\Delta^{1/3})^{|A|} + (d^{-(1/2-c)|A|})(d^{1/6}\Delta^{1/3})^{2|A|}$$

Here, the inequality follows the choice of $m = d^{1/6}\Delta^{-2/3}$. When $|A| \leq O(d^{c/4})$, we have

$$2^{-\Omega(d^c)}(d^{1/6}\Delta^{1/3})^{|A|} + (d^{-(1/2-c)|A|})(d^{1/6}\Delta^{1/3})^{2|A|}$$

$$= 2^{-\Omega(d^c)}(d^{1/6}\Delta^{1/3})^{|A|} + (d^{-1/6+c}\Delta^{2/3})^{|A|} \leq (d^{-1/6}\Delta^{2/3})^{|A|} = (1/100m)^{|A|}.$$

This implies that by the choice of $k = d^{c/4}$,

$$\underset{D^{(u)}, D^{(v)} \sim \mathcal{D}}{\mathbf{E}} \langle (D^{(u) \leq \infty, k}), (D^{(v) \leq \infty, k}) \rangle_{D^{(0)}} - 1$$

$$= \sum_{A \subseteq [m], A \neq \emptyset, |A| \leq k} \underset{D^{(u)}, D^{(v)} \sim \mathcal{D}}{\mathbf{E}} \prod_{j \in [A]} \left( \langle D_{T-m+j}^{(u)}(z_i)), D_{T-m+j}^{(v)}(z_i) \rangle_{D_i^{(0)}} - 1 \right) \leq \sum_{\ell=1}^{k} \binom{m}{\ell} (1/(100m))^\ell \leq 1.$$

$\square$

### D.5  From Testing to Learning

**Lemma D.6.** *Let $\mathcal{A}$ be any algorithm such that for any instance of learning $\gamma$-margin halfspaces under $\eta$-RCN it achieves error $\mathrm{opt}_i + \Delta^{1/3}\gamma^{-1/6}$ for $i \geq T$. Then $\mathcal{A}$ can be used to efficiently solve the instance of trajectory testing problem defined in Definition 4.3.*

*Proof of Lemma D.6.* We notice that by the choice of $m = d^{1/6}\Delta^{-2/3}$ and the design of $D_i^{(v)}$, we have $\mathbf{Pr}_{D_T^{(v)}}(h_T^{(v)}(x) = -1) = \Delta m = d^{1/6}\Delta^{1/3}$. Now, let $\hat{h}_T$ be the hypothesis output by $\mathcal{A}$ in the $T$-th round under $H_1$. We have

$$\mathrm{err}_T(\hat{h}_T) = \mathrm{opt}_T + (1 - 2\mathrm{opt}_T) \underset{D_T^{(v)}}{\mathbf{Pr}} (h_T^{(v)}(x) \neq \hat{h}_T) \leq \mathrm{opt}_T + d^{1/6}\Delta^{1/3}/100.$$

This implies $\mathbf{Pr}_{D_T^{(v)}}(h_T^{(v)}(x) \neq \hat{h}_T) \leq d^{1/6}\Delta^{1/3}/20$ and

$$\underset{D_T^{(v)}}{\mathbf{Pr}} (\hat{h}_T \neq 1) \geq \underset{D_T^{(v)}}{\mathbf{Pr}} (h_T^{(v)}(x) \neq 1) - \underset{D_T^{(v)}}{\mathbf{Pr}} (h_T^{(v)}(x) \neq \hat{h}_T) \geq 19d^{1/6}\Delta^{1/3}/20.$$

Similarly, under $H_0$, we have

$$\mathrm{err}_T(\hat{h}_T) = \eta + (1 - 2\eta) \underset{D_T^{(0)}}{\mathbf{Pr}} (\hat{h}_T \neq 1) \leq \eta + d^{1/6}\Delta^{1/3}/100,$$

which implies that $\mathbf{Pr}_{D_T^{(v)}}(\hat{h}_T \neq 1) \leq 3d^{1/6}\Delta^{1/3}/100$. This implies by running $\mathcal{A}$ over $z = (z_1, z_2, \ldots, z_T) \sim D$ and checking the probability of $\mathbf{Pr}_{D_T^{(v)}}(\hat{h}_T \neq 1)$, we are able to distinguish $H_0$ and $H_1$.

$\square$

To conclude Theorem 1.3, we make use of the low-degree polynomial hardness conjecture. Roughly speaking, the low-degree polynomial hardness conjecture says that for a natural high-dimensional statistical testing problem of size $n$, if there is

no polynomial of degree $O(\log n)$ that can solve the testing problem, then there is no polynomial-time algorithm that can solve it either. Originally stated in (Hopkins, 2018), variants of the conjecture have been used to give evidence of hardness for many problems, such as (Ding et al., 2021; Arpino & Venkataramanan, 2023; Ding et al., 2024). We summarize the conjecture informally below.

*Conjecture* 1. Let $\mathcal{P}_n$ and $\mathcal{Q}_n$ be two sequences of natural distributions of size $n$, if there is some $\epsilon > 0$ such that there is no $(\log n)^{1+\epsilon}$-degree polynomial as $n \to \infty$, which can distinguish $\mathcal{P}_n$ and $\mathcal{Q}_n$, then there is no polynomial time algorithm that can distinguish $\mathcal{P}_n$ and $\mathcal{Q}_n$ with probability $1 - o(1)$.

By Theorem 4.4, we know that there is no polynomial of degree less than $O(\gamma^{-1/8})$ that can solve the trajectory testing problem and thus there is no polynomial time algorithm that can distinguish the trajectory testing problem. Thus, Lemma D.6 implies that there is no polynomial-time algorithm that can solve the learning problem with error $\mathrm{opt}_T + \Delta^{1/3}\gamma^{-1/6}$.

