# OpenReview forum: "Efficiently Learning Drifting Halfspaces with Massart Noise"
_ICML.cc/2026/Conference — ICML 2026 regular_

### Official Review · Reviewer_sAA8 · 2026-03-03

**Soundness:** 3
**Presentation:** 3
**Significance:** 3
**Originality:** 3
**Overall Recommendation:** 5
**Confidence:** 4

**Summary:**

The paper considers the learning problem under distribution drift. They provide an efficient algorithm and characterize the error that can be achieved under the Massart noise assumption. Moreover, under the realizability assumption, their algorithm achieves a better error rate compared to prior results, and they proved that this error rate is optimal (in the minimax sense).

**Compliance With Llm Reviewing Policy:**

Affirmed.

**Final Justification:**

The paper is technically sound and benefits from weaker assumptions than prior work, which strengthens its contribution. The rebuttal satisfactorily addresses my main concerns, and these clarifications improve my assessment of the paper’s clarity and significance.

**Key Questions For Authors:**

Can your algorithm be easily extended to an adaptive version that does not require knowledge of the Massart parameter $\eta$ and the drift rate $\Delta$? if so, how?

In Theorem 1.2, the drift and the noise parameters contribute separately (additive). Is this separation fundamental? or is there a regime where the interaction between drift and noise fundamentally changes the achievable error rate?

**Limitations:**

Yes

**Strengths And Weaknesses:**

Soundness: The paper appears technically sound, with claims that are well supported by theoretical analysis. The proofs seem correct and the assumptions seem reasonnable.

Presentation:  The paper is generally well written. The author can formally explain what $\gamma$-margin halspace means (one sentence is enough). The assumptions are not discussed sufficiently, for instance it is not clear from the text whether the $\gamma$- margin assumption is related to prior assumptions (Hanneke et al 2015, Crammer et al 2010).

Significance: The work addresses a relevant problem in learning under distribution drift, both from algorithmic and statistical point of view.

Originality: The work provides new theoretical insights into learning under distribution shift under mild distributional assuptions. This seems new to me and represents a meaningful contribution.

---

> ### Author Rebuttal · Authors · 2026-03-31
>
> # Response to Reviewer sAA8:
> We would like to thank the reviewer for appreciating our work and acknowledging the importance of our results. We respond to the reviewer on the perceived weakness and related questions.
>
> >The assumptions are not discussed sufficiently, for instance it is not clear from the text whether the -margin assumption is related to prior assumptions (Hanneke et al 2015, Crammer et al 2010).
>
>
> Hanneke et al. (2015) assumes that the marginal distribution is the uniform distribution on the unit sphere, and Crammer et al. (2010) assumes a $\lambda$-good distribution, which can be viewed as a more general global regularity condition motivated by properties of the uniform spherical distribution. These assumptions are significantly stronger than the margin assumption we consider. Specifically, such strong assumptions enable prior works to show that the classification error between two halfspaces is proportional to their angle, which implies that a very small fraction of examples have a small margin.
> In contrast, our paper makes a substantially weaker assumption in terms of global distributional structure: we do not impose any specific shape condition on the marginal distribution but only require that the distribution places no mass in a narrow band around the halfspaces.
>
>
> >Can your algorithm be easily extended to an adaptive version that does not require knowledge of the Massart parameter and the drift rate ? If so, how?
>
>
> **Adaptivity to Parameter $\Delta,\eta$.** As most prior works in the learning theory literature on this topic (Helmbold et al., 1991; Bartlett, 1992; Helmbold & Long, 1994; Barve & Long, 1997; Long, 1999), our algorithm makes the simplifying assumption that the parameters $\Delta$, $\eta$ are known.
> We remark that there is a standard way to deal with the case when one or two input parameters are unknown. That is to say, we can guess these input parameters by doubling the guess in each round, run the learning algorithm with the guess multiple times, and use hypothesis testing to select a good hypothesis. This will only take logarithmically many rounds and only introduce a logarithmic factor in the final error.
>
> >In Theorem 1.2, the drift and the noise parameters contribute separately (additive). Is this separation fundamental? or is there a regime where the interaction between drift and noise fundamentally changes the achievable error rate?
>
> In the noise setting, the way to express the final error is $opt$+drift error, which is information theoretically unavoidable. In fact, the algorithm under RCN and realizable setting is exactly stated in this way. For the more general case of Massart noise, we instead state it as $\eta$+drift error. This is unavoidable for efficient learning algorithms. As we mentioned in the introduction, even without distribution drift, it is computationally hard to achieve an error below $\eta$. This fact was also mentioned in lines 162,163 in the introduction.
>
> >Hanneke, S., Kanade, V., and Yang, L. Learning with a drifting target concept. In International Conference on Algorithmic Learning Theory, pp. 149–164, 2015.
>
> >Crammer, K., Mansour, Y., Even-Dar, E., and Vaughan, J. W. Regret minimization with concept drift. In COLT, pp. 168–180, 2010.

---

> > ### Author Rebuttal · Reviewer_sAA8 · 2026-04-03
> >
> > Thank you to the authors for addressing my concerns. I appreciate your response and raise my score to 5.

---

> > > ### Author Response · Authors · 2026-04-03
> > >
> > > Thank you for your thoughtful review and for your positive feedback on our revision. We sincerely appreciate your time, effort, and support, and we are especially grateful for your change of recommendation to accept.

---

### Official Review · Reviewer_yTRa · 2026-03-10

**Soundness:** 3
**Presentation:** 3
**Significance:** 3
**Originality:** 3
**Overall Recommendation:** 4
**Confidence:** 4

**Summary:**

This paper studies online learning with drift under Massart noise. The target is a margin halfspace, and adjacent distributions differ by at most $\Delta$ in total variation. The paper gives three main results:

* an information-theoretic upper bound of about $\widetilde O(\sqrt{\Delta/\gamma})$,
* a polynomial-time algorithm with excess error about $\widetilde O(\Delta^{1/3}/\gamma)$,
* low-degree hardness evidence suggesting this efficient rate may be hard to improve.

The main technical idea is a regret-to-error lemma that avoids directly tracking how the target separator moves geometrically. Overall, the paper has a clear and interesting high-level story: there may be a real gap between the statistical rate and the efficient rate in this drifting noisy setting.

**Compliance With Llm Reviewing Policy:**

Affirmed.

**Final Justification:**

The rebuttal satisfactorily addressed my main technical concern. In particular, the authors provided a simpler and convincing argument showing that the optimal error changes by at most $\Delta$ between consecutive rounds, which resolves the key proof issue I had raised around the RCN extension and makes the discretization step credible. The authors also gave a more concrete justification for the modeling assumptions, clarifying why the margin and Massart-noise assumptions are standard and meaningful in the context of a theory paper. While some limitations remain, especially that the setting is still somewhat idealized and questions such as adaptivity to unknown drift or softer margin assumptions are left open, but I now view these as scope limitations rather than correctness problems. Given the novelty of the results, the fairly complete theoretical picture, and the clarification provided in the rebuttal, I am updating my recommendation to weak accept.

**Key Questions For Authors:**

* Can the authors provide a corrected proof of Theorem 3.6?

**Limitations:**

* The method appears to depend on knowing or tuning to $\Delta$. The paper does not discuss adaptation when $\Delta$ is unknown.
* The guarantees depend strongly on the margin parameter $\gamma$, but the paper does not discuss approximate or soft margin settings.

**Strengths And Weaknesses:**

Strengths:

* The problem is natural and important.
* The paper gives a fairly complete picture: statistical upper bound, efficient algorithm, and hardness evidence.
* The main proof idea is interesting and seems novel.
* The efficient guarantee under Massart noise appears nontrivial.

Weaknesses:

* There seems to be a real proof error in Theorem 3.6. The proof uses a displayed inequality of the form: $\mathrm{opt}\_{t+1} + (1-2\mathrm{opt}\_{t+1}) \Pr(\text{disagreement}) \le \Delta.$ As written, this cannot be true in general, since the left side contains $\mathrm{opt}_{t+1}$ plus a nonnegative term. The earlier claim only bounds the disagreement probability by an $O(\Delta)$-type quantity. It does not imply this full inequality.
* The practical value of the setting is not fully convincing. The theory setting is clean, but it is somewhat narrow from a practical view. The paper assumes all of the following at once: the target is always a halfspace, there is a uniform margin parameter $\gamma$, drift is bounded in adjacent total variation distance, and labels satisfy Massart noise. Each assumption is standard on its own, but together they define a fairly idealized regime. The paper does not give much discussion of where this exact combination arises in practice, or why this is the right model for realistic drifting data.

---

> ### Author Rebuttal · Authors · 2026-03-31
>
> # Response to Reviewer yTRa:
>
> We would like to thank the reviewer for carefully reading our paper and for pointing out a typo in the proof.
>
> ## Correctness:
> We want to thank the reviewer for pointing out this typo.
>
> First, we note that the statement of the theorem is correct; this is because we are not interested in optimizing the constant hidden in tilde $O(\Delta^{1/3}/\gamma)$.
> Second, the $opt_{t+1}$ term is a typo and should not appear before $(1-2opt_{t+1})Pr(\text{disagreement region})$.
> Third, since we do not care about the absolute constant hidden in the big O notation, what the proof actually does is to show there is some $\eta$, such that $\eta-c \Delta T<opt<\eta$, so that when we select a rough guess of $\eta$, the algorithm under Massart noise case still works, which requires to show that $|opt_{t+1}-opt_t|<O(\Delta)$. So, even if there is a hidden constant in the earlier claim, it does not affect the overall correctness of the proof, so long as the constant is matched carefully.
> We acknowledge that the hidden constants were not dealt with carefully enough. We appreciate the reviewer for drawing our attention to this issue, which we will fix in a revision.
>
> ## Practical Significance:
> This is a theory paper on a well-studied problem in the literature, thus well-within the scope of the conference, as per the current call for papers.
> Regarding the concern about practical value, we would like to respond to the reviewer from two perspectives. First, the motivation for studying the question is to provide a deeper understanding of the computational efficiency of learning problems under distribution drift. The problem of halfspace learning under distribution drift is a canonical problem in this direction that provides foundations for designing algorithms that work under broader families of learning problems such as generalized linear models, or simple neural networks. Second, as we mentioned in the introduction, prior works that design efficient learning algorithms have mostly focused on idealized settings, where the distribution is a uniform distribution over a sphere with no label noise. That is to say, the setting studied by this paper has already significantly generalized the setting studied by prior work, both in the marginal distribution and the label noise.
>
>
> We next answer questions from the reviewer.
>
> ## Adaptivity to Parameter $\Delta$.
>
> As most prior works in the learning theory literature on this topic (Helmbold et al., 199; Bartlett, 1992; Helmbold & Long, 1994; Barve & Long, 1997; Long, 1999
> ), our algorithm makes the simplifying assumption that the parameter $\Delta$ is known.
> We remark that there is a standard way to deal with the case when one or two input parameters are unknown. That is to say, we can guess these input parameters by doubling the guess in each round, run the learning algorithm with the guess multiple times, and use hypothesis testing to select a good hypothesis. This will only take logarithmically many rounds and only introduce a logarithmic factor in the final error. On the other hand, the design of an algorithm that achieves a data-driven error guarantee, i.e., so that the error is a function of the input parameter $\Delta_i$,  requires a more dynamic way to design the window size for the learning problem (see, e.g., Mazzetto and Upfal (2023). That said, this is beyond the scope of our work, whose central message is the existence of an information-computation tradeoff.
>
>
>
> ## Margin Assumption
> While it would be interesting to consider simple generalizations of the margin assumption (e.g., soft margin), the task of learning halfspaces with a margin is one of the most basic tasks in machine learning—starting with the Perceptron algorithm. Moreover, in the offline learning setting, there is a standard method to reduce the distribution-free setting to the large margin setting, via a technical tool known as Forster’s transform (Diakonikolas et al. 2023). That said, this approach cannot be directly applied to the drift setting for a number of technical reasons. Developing a drift algorithm for the distribution-free setting for halfspaces is left as an interesting open problem.
>
> >Diakonikolas, Ilias, Daniel Kane, and Christos Tzamos. "Forster decomposition and learning halfspaces with noise." Advances in Neural Information Processing Systems 34 (2021): 7732-7744.
>
> >Mazzetto, A. and Upfal, E. An adaptive algorithm for learning with unknown distribution drift. Advances in Neural Information Processing Systems, pp. 10068–10087, 2023.

---

> > ### Author Rebuttal · Reviewer_yTRa · 2026-04-03
> >
> > Thank you for the clarification. I appreciate the authors’ explanation that the problematic line in the proof of Theorem 3.6 is a typo and that the intended argument is only to bound the disagreement contribution, so as to show that $\mathrm{opt}_t$ changes by $O(\Delta)$ within an epoch. This makes the intended proof strategy clearer.
> >
> > However, I do not think the correctness concern is fully resolved in the current rebuttal, because the response does not provide a corrected inequality chain or the necessary constant bookkeeping. Since the RCN adaptation subsequently relies on discretizing $\eta$, these constants do matter. On the practical-significance side, the response clarifies the theoretical motivation and the relation to prior, more idealized settings, but it still does not fully explain why this exact combination of assumptions is the right model for realistic drifting data.

---

> > > ### Author Response · Authors · 2026-04-03
> > >
> > > We thank the reviewer for engaging in a discussion and giving us the opportunity to address any remaining concerns.
> > >
> > >
> > > We first provide a detailed fix of the issue mentioned by the reviewer.
> > >
> > > We would like to clarify that the constant does not affect the discretization step. Suppose there is a constant $c>0$ such that $|opt_i - opt_{i+1}| \le c \Delta$.
> > > Then there must be some $\eta$ such that for every $i$, we have $\eta-c \Delta T\le opt_{t+i}\le\eta$. This implies that if we discretize the interval [0,1/2] with a grid size $\Delta T/2$, it must be the case that we have to get such an $\eta$ such that for every $i$,
> > > $\eta - (c+1/2) \Delta T\le opt_{t+i}\le\eta$. Now, with this value of $\eta$, we get $err^{(t)}(h^{(t)}))\le \eta+ \tilde O(\Delta^{1/3}/\gamma)$. Notice that the distance between $\eta$ and $opt$ is at most $(c+1/2)\Delta T \le (c+1)\tilde O(\Delta^{1/3})$, which is a low-order term compared to the excess error term. So the statement is still correct.
> > >
> > > To address the reviewer’s comment, we argue that $c=1$ suffices, by using an even simpler proof, without invoking Claim 3.3. Notice that $err^{(t+1)}((w^*)^{(t)}) = opt_{t+1} + (1-2opt_{t+1})\Pr(disagreement)\ge opt_{t+1}$
> > >
> > > This implies that $opt_{t+1}\le opt_t+\Delta$; otherwise, $err^{(t+1)}((w*)^{(t)}) > err^{(t)}((w*)^{(t)})+\Delta$
> > > But notice that the roles of $t+1$ and $t$ are symmetric, which means $opt_{t}\le opt_{t+1}+\Delta$ also holds.
> > > This gives that $|opt_t - opt_{t+1}|\le \Delta$.
> > >
> > >
> > > We next address the practical significance of our assumptions. The “large margin” assumption essentially means that the distribution on the feature space can be arbitrary, the only constraint being that no points lie very close to the target linear classifier (i.e., there is a target classifier that “reliably” classifies the ground-truth data, without the label noise). As we mentioned in our earlier response, this is one of the standard assumptions in machine learning, going back to the Perceptron algorithm. Specifically, it allows one to use Kernel methods to handle much broader and more expressive concept classes beyond linear classifiers.
> > > Moreover, even in the kernel setting, Massart noise is the prototypical model of semi-random label noise that is widely used for modeling label corruptions arising from a range of practical applications, such as corruption during label collections or model misspecifications. Notably, handling fully adversarial label noise for margin halfspaces is known to be computationally intractable even in the easier offline setting (as noted in lines 155-157). So, the Massart noise model is essentially the strongest noise model where there is a hope for a polynomial time algorithm in our setting (and we obtain such an algorithm). As a summary, under proper feature engineering, the model studied in this paper can be used to capture a wide family of classes that are expressive and evolve as time changes, while the Massart noise assumption naturally captures the label corruption arising from data collection or model misspecification.

---

### Official Review · Reviewer_hysv · 2026-03-11

**Soundness:** 3
**Presentation:** 2
**Significance:** 2
**Originality:** 3
**Overall Recommendation:** 4
**Confidence:** 3

**Summary:**

This paper studies the problem of learning margin halfspaces under distribution drift and Massart label noise. It focuses on the online setting: at each time step, the learner must output a predictor and then observes a sample from the current data distribution. The paper aims to bound the predictive error of the learner at each time step uniformly, for all sufficiently large time steps. The paper makes three technical contributions:

1. It establishes an information-theoretic upper bound of $\tilde{O}(\sqrt{d\Delta})$ on the error under Massart noise, and proves a matching lower bound in the simpler RCN setting.
2. It gives a polynomial-time algorithm achieving an upper bound of $\tilde{O}(\Delta^{1/3})$ on the excess error.
3. It introduces a hypothesis testing problem that can be solved efficiently if the drifting learning problem can be solved efficiently. The paper then shows that this testing problem cannot be solved by low-degree polynomials, providing evidence for a computational barrier.

**Compliance With Llm Reviewing Policy:**

Affirmed.

**Final Justification:**

In light of the rebuttal, I am increasing my score to 4, conditional on the authors toning down the implications of their computational lower-bound claim in the revision.

I remain unconvinced by the rebuttal’s argument that a low-degree polynomial learner would imply a low-degree polynomial tester. As written, the reduction seems to establish only an efficient distinguisher, not a low-degree one, and it is not shown that the reduction preserves degree. This preservation is important for the stated low-degree hardness implication. If the authors wish to keep this stronger interpretation (for example, by suggesting that the learning problem cannot be solved by SQ algorithms) they should make this argument fully rigorous in the revision.

**Key Questions For Authors:**

See weaknesses section.

**Strengths And Weaknesses:**

**Strengths:**

1. The problem of efficiently learning halfspaces under Massart noise with distribution drift appears to be a natural and interesting theoretical problem.
2. The paper provides a collection of non-trivial excess-risk bounds, both from the information-theoretic side and from the algorithmic side.
3. The paper also provides an upper bound for the realizable case that appears to improve upon the earlier bound of Helmbold & Long (1994).

**Weaknesses:**

While this is certainly a technically solid paper, I find the contribution relatively incremental. To me, the paper mainly combines several well-studied ingredients into a setting that has not been studied before. That can still make for a strong paper if the resulting solution introduces a sufficiently novel conceptual contribution, but I am not convinced that this is the case here, for the following reasons:

1. The main algorithmic contribution (Algorithm 2) seems to rely heavily on the “regret-to-error” reduction introduced in Diakonikolas et al. (2025), with the main new ingredient being the additional handling of distribution drift.
2. The computational lower bound is overstated. What the paper actually shows is that a related testing problem cannot be distinguished by low-degree polynomial testers. This is much weaker than what the paper sometimes seems to suggest, namely a genuine information-computation gap for the original learning problem. In fact, the framing initially misled me into thinking that the authors had proved a more direct computational hardness result.
3. The information-theoretic upper bound also seems to follow largely by adapting existing techniques to the Massart-noise setting, rather than introducing genuinely new conceptual ideas.

**Overall:**

While I do believe this is a technically solid paper, I find the current contribution too incremental for ICML. I am therefore leaning toward a *weak reject*.

---

> ### Author Rebuttal · Authors · 2026-03-31
>
> # Response to Reviewer hysv:
>
> We would like to thank the reviewer for the time and effort in reviewing our paper. We respectfully disagree with the provided summary and interpretation of our results, and address the specific points below after we summarize our contributions.
>
> ## Summary of Contributions:
>
> We start by pointing out that, although we give a **tight** characterization of the information-theoretical optimal error for this problem, the statistical complexity of the problem is not the focus of our paper. That said, the information-theoretic limits are new and are necessary to contrast with our algorithmic and lower bound guarantees, and establish an information-computation tradeoff (which is the main contribution of this work).
>
>
> Our first main contribution (Theorem 1.2) is to provide **the first computationally efficient learning algorithm** for margin halfspaces under distribution-drift and in the presence of **Massart noise** with **near-optimal error guarantees**. Moreover our algorithm succeeds under much milder distributional assumptions compared to prior work. For comparison, works cited as Crammer et al. (2010) and Hanneke et al. (2015) in our paper work under the **uniform distribution on the sphere** (a much stronger distributional assumption) and only in the **realizable** setting.  In fact, our algorithm even provides a new result for the realizable setting with a margin (see Theorem 3.7): this gives the first non-trivial improvement in the error guarantee over the linear programming method ($\sqrt{\Delta}\gamma^{-2}$ v.s. $\sqrt{\Delta}\gamma^{-3/2}$ ).
>
> Our second main contribution (Theorem 4.4) is a lower bound in the low-degree polynomial (LDP) framework, giving rigorous evidence that the error guarantee achieved by our algorithm is near-optimal within the class of polynomial-time algorithms. To prove this result, we give a new methodology to establish LDP lower bounds for online learning problems that is novel, both conceptually and technically.
>
>
> ## Response on Technical Novelty:
>
> We now comment on the technical novelty of our algorithm and its analysis. First, we emphasize that the idea of using online learning ("regret" bounds) as a tool for offline/passive learning has been leveraged in numerous works in the literature. The conceptual difficulty in the drift setting is in designing the “correct” optimization algorithm that when balanced with the drift error leads to a near-optimal result. As a clear indication of the novelty of our ideas, we remark our improved upper bounds for the realizable setting, as compared to vanilla linear programming Helmbold & Long (1994). Both our algorithm for the drift setting and its analysis are fundamentally different from the setting and analysis of Diakonikolas et al. (2025). Conceptually, there is no reduction from the setting of Diakonikolas et al. (2025) to our online drifted setting. This is because the target concept drifts in each time step, and therefore the examples behave as being subject to not only Massart noise but also to *adversarial* noise due to drift. What this means for the analysis, for instance, is that we need to carefully bound the error contribution of examples both near and far from the hypothesis in a different manner and balance it with the drift error—which is not reflected in the analysis carried out in prior work.
>
>
> Regarding our information-computation tradeoffs: First, it is a standard approach in the literature to establish information-computation tradeoffs for learning (search) problems by (1) defining an appropriate hypothesis testing problem that reduces to learning (i.e., if the testing task is “hard”, so is the learning task), and (2) proving a lower bound for the testing task. That is to say the way we phrase our computational hardness result is standard.
> Moreover, the LDP (low-degree polynomial) testing framework is a well-established model of computation: hardness in this model is accepted as strong evidence of computational hardness, as the LDP model captures almost a wide range of algorithms, such as gradient-descent based methods, spectral methods, and moment methods (see e.g. the survey Wein (2025)). Using this well-established recipe, we prove an LDP lower bound for learning margin halfspaces in the drift setting, providing formal evidence that there is no efficient learning algorithm that can reach the statistical limit. Moreover, even formalizing the appropriate testing problem for the (online) drift setting is challenging and a novel conceptual and technical contribution of our work.
>
> >Diakonikolas,I.,Kontonis,V.,Tzamos,C.,andZarifis,N.Online linear classification with Massart noise. In International Conference on Machine Learning(ICML), 2025
>
> >Helmbold,D.P.andLong,P.M.Tracking drifting concepts by minimizing disagreements. Machinelearning,14(1):27–45, 1994
>
> > Wein,A.S.Computational complexity of statistics:New insights from low-degree polynomials, 2025

---

> > ### Author Rebuttal · Reviewer_hysv · 2026-04-02
> >
> > I read the authors’ rebuttal, which clarified some of my concerns, and I intend to increase my score to 4. However, I still maintain my position that the computational lower-bound result is overstated. In particular, the paper explicitly states:
> >
> > > “An intuitive interpretation of Theorem 1.3 is that efficient algorithms for our problem require excess error \(\Omega(\Delta^{1/3})\), matching the guarantee of our algorithm.”
> >
> > I think this is misleading. As Wein (2025) puts it, the low-degree framework “yields reliable conjectures about inherent computational hardness,” but one should also be “transparent about many caveats” and readers should be “not … overconfident about these conjectures.”
> >
> > Moreover, the rebuttal seems to suggest that the low-degree testing lower bound rules out algorithms such as SGD for the original learning problem, rather than only for the associated testing problem together with a reduction. Can the authors point to a precise theorem establishing this?
> >
> > I am not saying that reducing to a low-degree testing problem is unacceptable; rather, my concern is that the paper currently overstates what this type of result actually establishes.

---

> > > ### Author Response · Authors · 2026-04-03
> > >
> > > We thank the reviewer for engaging in a discussion and for adjusting the score based on our first response. We now understand better what the reviewer meant.
> > >
> > > Regarding the description of our LDP testing lower bound. While an LDP testing lower bound provides evidence of hardness, it is of course “possible” that there exists a polynomial-time algorithm that is not captured by the underlying model of computation. At the end of the day, LDP is a broad yet restricted computational model. This is why the phrasing that we used in the abstract of our paper is that “we provide formal evidence” of an information-computation tradeoff, and “suggesting that”. The sentence cited by the reviewer starts with “intuitive interpretation”. While such a phrasing is fairly standard in a range of papers establishing the LDP lower bound, we would be happy to rephrase in order to clarify any confusion.
> > >
> > > In summary, LDP still covers a broad range of algorithms, including all SQ algorithms (which can be used to simulate GD) under natural assumptions. See the discussion in Section 7.3 of Wein’s survey and Brennan et. al.(2021).
> > >
> > > The reviewer is correct that our result establishes LDP lower bound for the testing problem, which is by definition computationally easier than the learning problem.  We note that this type of approach is standard in a wide range of papers in this field. That is, if there is a low-degree polynomial learning algorithm, then this would imply a low-degree polynomial tester, which leads to a contradiction.
> > >
> > > >Matthew S Brennan, Guy Bresler, Sam Hopkins, Jerry Li, and Tselil Schramm. Statistical query algorithms and low degree tests are almost equivalent. In Conference on Learning Theory, pages 774–774. PMLR, 2021

---

### Official Review · Reviewer_JAwH · 2026-03-12

**Soundness:** 3
**Presentation:** 4
**Significance:** 3
**Originality:** 3
**Overall Recommendation:** 4
**Confidence:** 2

**Summary:**

In this paper, the authors investigate the problem of how to efficiently learn drifted halfspaces with margin under the Massart noise condition. The problem is as follows. The learning procedure is sequential, there are several different distributions indexed by $t = 1,2,3, \ldots$. The total distance between consecutive two distributions is smaller than $\Delta$. And each distribution satisfies the Massart noise condition with respect to the concept class $\mathcal{H}$, which is a class of halfspaces with margin $\gamma$, with parameter $\eta$. The goal is to minimize the population error of the output hypothesis at round $T$, when $T$ is large enough. They provide an efficient algorithm that gets an error rate of $\eta+\tilde{O}(\Delta^{1/3}/\gamma)$. On the other hand, they show a matched lower bound when we require that algprothm to be efficient.

**Compliance With Llm Reviewing Policy:**

Affirmed.

**Final Justification:**

My concerns have been fully resolved, and I will keep my positive assessment.

**Key Questions For Authors:**

Is it possible to investigate the computational efficiency problem on other concept classes, such as neural nets?
Will there be similar computational-statistical trade-offs on other learning problems?

**Limitations:**

Yes

**Strengths And Weaknesses:**

Computational efficiency is very important topic in the field of theoretical computer science. It is very important to understand the efficiency of learning algorithms and the trade-off between the statistical aspects (error rate) and the computational aspects (running time). However, only several learning problem are provablly efficiently solvable, such as learning halfspaces with margin. In this paper, the authors follows this line and asks the computational efficiency of learning drifted halfspaces and provide a polynomial algorithm with error rate $\eta+\tilde{O}(\Delta^{1/3}/\gamma)$. They further accomplish this result with a lower bound and combining with the error rate reached by ERM (which is not efficiently computable). They also show that in this problem, there is a trade-off between computational efficiency and sample efficiency. The results are interesting and strong. The papers are well written.
The weakness I can think is that efficiently learning halfspaces is a little bit less general and there has been many works on this topic. If the results are for other settings, it will be much more impressive.

---

> ### Author Rebuttal · Authors · 2026-03-31
>
> # Response to Reviewer JAwH:
>
> We would like to thank the reviewer for appreciating our work and acknowledging the importance of our results. Below, we address the perceived potential weakness and related questions.
>
> >learning halfspaces is a little bit less general and there has been many works on this topic. If the results are for other settings, it will be much more impressive.
>
> We would like to clarify that although the problem of learning halfspaces has been studied in various settings, under the problem of learning with distribution drift, the computational efficiency of this problem is still very poorly understood (although it has been studied since the 1990s). For example, even without label noise, the result in our paper (Theorem 3.7) gives the first non-trivial improvement beyond the linear programming approach in the literature. That is, even in that simple setting, there was still a non-trivial gap from the information-theoretic optimal error rate standpoint. This suggests that a good understanding of this fundamental problem plays a fundamental role in understanding the complexity landscape for more general classes. For example, the highly expressive kernel method relies on algorithms for learning halfspaces. Prior to our work, there was no known methodology to characterize the computational possibilities and limitations of this problem, given its online nature. To the best of our knowledge, our work is the first one that provides a comprehensive picture of the computational aspects of this fundamental problem, and thus also builds the foundations for more general problem classes in the online/drift settings.
>
>
> > Is it possible to investigate the computational efficiency problem on other concept classes, such as neural nets? Will there be similar computational-statistical trade-offs on other learning problems?
>
>
> We would like to thank the reviewer for pointing out these potential directions; we think they are important and promising to study as natural follow-up works in this line of research. To the best of our knowledge, the computational aspects of learning with a drift have mostly been studied for classification problems. Extending our understanding to real-valued functions, such as generalized linear models or more general single-index models is a natural open question. We conjecture that gradient descent-style updates would similarly lead to efficient algorithms for learning such functions under distribution drift and that similar information-computation trade-offs would also exist. That said, the study of these questions is beyond the scope of the present work. For more general, large neural networks, the computational complexity of learning is not yet fully understood even in the vanilla PAC setting (i.e., without label noise or distribution drift).

---

> > ### Author Rebuttal · Reviewer_JAwH · 2026-04-03
> >
> > I thank the authors for their responses; my concerns have been fully resolved and I will keep my positive score.

---

> > > ### Author Response · Authors · 2026-04-03
> > >
> > > Thank you for your thoughtful review and for your positive feedback on our revision. We sincerely appreciate your time, effort, and support.

---

### Decision · Program_Chairs · 2026-04-30

**Decision:**

Accept (regular)

**Comment:**

The paper studies learning a "drifting concept" under the Massart noise model, giving a new algorithm as well as complementary hardness evidence for low-degree polynomials. The reviewers were generally optimistic about the cleanliness of presentation and the relevance of the problem considered; there were a few concerns about the theory that were addressed to the reviewers' satisfaction during the rebuttal phase. On the other hand, the main limitations discussed were the somewhat restrictive problem under consideration (as opposed to general theory beyond halfspaces) and an overstatement of the strength of the lower bound. Overall, I think the paper is of good quality and support acceptance, but I request that the authors take into consideration the reviewers' points of confusion and address them in a revision.